# Increased reactivity of the paraventricular nucleus of the hypothalamus and decreased threat responding in male rats following psilocin administration

Devin P. Effinger [1,2], Jessica L. Hoffman[2], Sarah E. Mott[1,2], Sarah N. Magee[2], Sema G. Quadir [2], Christian S. Rollison[2], Daniel Toedt[2], Maria Echeveste Sanchez[2], Margaret W. High[2], Clyde W. Hodge[2] & Melissa A. Herman [1,2] ✉

Psychedelics have experienced renewed interest following positive clinical effects, however the neurobiological mechanisms underlying effects remain unclear. The paraventricular nucleus of the hypothalamus (PVN) plays an integral role in stress response, autonomic function, social behavior, and other affective processes. We investigated the effect of psilocin, the psychoactive metabolite of psilocybin, on PVN reactivity in Sprague Dawley rats. Psilocin increased stimulus-independent PVN activity as measured by c-Fos expression in male and female rats. Psilocin increased PVN reactivity to an aversive air-puff stimulus in males but not females. Reactivity was restored at 2- and 7-days post-injection with no group differences. Additionally, prior psilocin injection did not affect PVN reactivity following acute restraint stress. Experimental groups sub-classified by baseline threat responding indicate that increased male PVN reactivity is driven by active threat responders. These findings identify the PVN as a significant site of psychedelic drug action with implications for threat responding behavior.

Clinical studies suggest that psychedelics may provide rapid-acting and long-lasting improvements in psychiatric conditions including treatment-resistant depression (TRD), alcohol use disorder (AUD), and smoking cessation[1–10]. Psilocybin is currently in phase III clinical trials for TRD[11–13], and symptom improvements have been observed after a single dose[12,13]. Psilocybin is dephosphorylated into its active metabolite psilocin, which crosses the blood-brain barrier[14] to evoke alterations in synaptic structure and function[15–17]. The 5-hydroxytryptamine 2 A receptor (5-HT$_{2AR}$) has been shown to be a primary receptor mediating the hallucinogenic effects of psychedelics[17–20]. However, recent evidence suggests that the antidepressant[21,22] and synaptogenic effects[15] may be independent of 5-HT$_{2AR}$ activation, with recent work

implicating the BDNF TrkB receptor[22]. Despite advancements in knowledge, the neurobiological mechanisms underlying the fast-acting and long-lasting effects of these drugs remain unclear.

Psychiatric disorders including post-traumatic stress disorder (PTSD), anxiety, and major depressive disorder (MDD) are associated with dysregulation of the hypothalamic-pituitary-adrenal axis (HPA axis)[23–26], which orchestrates the central and peripheral stress response[27,28]. The paraventricular nucleus of the hypothalamus (PVN)[29] is an integral hub within the HPA axis but has also been implicated in a wide array of affective, social, and stress-related behavioral responses. To investigate the effects of psilocin on PVN function, we utilized histology, fiber photometry (FP), and simultaneous behavioral

[1]Department of Pharmacology, University of North Carolina at Chapel Hill, Chapel Hill, NC, USA. [2]Bowles Center for Alcohol Studies, University of North Carolina at Chapel Hill, Chapel Hill, NC, USA. ✉e-mail: melissa_herman@unc.edu

recording to measure changes in PVN activity and the corresponding behavioral response to an aversive stimulus. Additionally, we performed restraint stress to examine how prior psilocin exposure altered PVN reactivity following an acute stressor. Here, we provide findings pairing PVN reactivity with behavioral responding to an aversive air-puff stimulus and demonstrate sex differences in baseline responding. Additionally, we test how the serotonergic psychedelic, psilocin, alters PVN activity, highlighting sex-specific effects of psilocin administration on stimulus-evoked PVN reactivity and behavior.

## Results

### Increased activity within the paraventricular nucleus of the hypothalamus (PVN) following psilocin administration

Male and female Sprague Dawley rats (~8 weeks old) were injected with vehicle or psilocin (2 mg/kg, s.c.) 2 hr prior to perfusion. Immunohistochemical analysis and quantification of PVN c-Fos expression as a marker of basal neuronal activation was performed. As injections are inherently stressors, vehicle control groups are included to account for any injection-related confounds, and c-Fos+ expression in both groups may be elevated compared to basal conditions. Following analysis of coronal sections including the PVN, we found that psilocin administration significantly increased c-Fos expression in the PVN of males ($t(8) = 2.603$, $p = 0.03$; Fig. 1A) and females($t(8) = 2.324$,$p = 0.04$; Fig. 1B) compared to vehicle control, suggesting that psilocin administration resulted in increased stimulus-independent activity in the PVN for both males and females.

### Acute stimulus-evoked effects of psilocin on PVN reactivity and behavior

Male and female Sprague Dawley rats (~8 weeks old, 250-300 g, Fig. 1C) received microinjections of the genetically encoded calcium sensor pGP-AAV-syn-jGCaMP7f-WPRE followed by fiber optic implantation into the PVN (Fig. 1C). Group sizes were determined based on previous work[30]. After 3 weeks to allow for viral transfection and recovery, fiber photometry recordings were performed in conjunction with exposure to an air-puff stimulus (Figs. 1C, D). Baseline recordings were conducted to determine PVN reactivity prior to drug administration. Animals were assigned to groups following baseline recording sessions to ensure that reactivity between groups was consistent at baseline, thus accounting for any potential differences in viral transfection and GCaMP expression between groups. Following baseline recordings, animals received administration of vehicle or psilocin (2 mg/kg, s.c.). Follow-up recordings were performed 2- and 6/7-days post-injection. On day 8, recordings were performed after a 20-minute restraint stress to test for the effects of an acute stressor on PVN reactivity and the potential effects of prior psilocin administration. To assess changes in individual threat responding, maximum speed values (velocity) and distance traveled were collected within the 10 sec period following air-puff to align immediate behavioral responding with PVN reactivity. Following the completion of all air-puff fiber photometry experiments, brain tissue was collected, and site verification was conducted to ensure proper virus infection and fiber placement (Fig. 1E). At baseline, both males and females exhibited increases in PVN reactivity in response to the air-puff stimulus. However, PVN reactivity in the males was significantly greater than in the females (2-way ANOVA: $F_{interaction}(19,608) = 5.646$, $p < 0.0001$, Fig. 1F), with a significantly greater increase in Peak Point (PP, $t(32) = 2.406$, $p = 0.02$). Therefore, males and females were split into separate groups for all analyses moving forward.

In males, no differences were observed between groups in locomotor behavior or time spent in the center of the open field box during habituation or the inter-stimulus interval (ISI) period (Fig. S1A). Following baseline air-puff, there was a significant increase in PVN reactivity, with no differences seen in Area under the Curve (AUC) or PP between vehicle and psilocin groups (Fig. 2A). There was a significant

interaction in baseline max speed following air-puff, with a greater increase in velocity seen in vehicle control animals at 2 seconds compared to the psilocin group (2-way ANOVA: $F_{interaction}(9189) = 2.444$, $p = 0.01$, Fig. 2B). All animals showed an increase in distance traveled following air-puff, but no differences were seen between groups (Fig. 2B). On the day of injection, male subjects received either vehicle or psilocin (2 mg/kg, s.c.) injection and were placed in their home cage for 30 minutes prior to being connected to the fiber photometry system and placed into the open field box for video-monitored habituation. Following injection, psilocin treated animals showed significant reductions in distance traveled during the habituation period (2-way ANOVA: $F_{time}(14,224) = 15.65$, $p < 0.0001$; $F_{interaction}(14,224) = 1.795$, $p = 0.04$; $F_{treatment}(1,16) = 13.41$, $p = 0.002$; Fig. S1B). Additionally, psilocin treatment significantly reduced time spent in the center as compared to vehicle control, with significant differences seen at 4 and 5 minutes (2-way ANOVA: $F_{interaction}(14,224) = 3.160$, $p = 0.0002$; Fig. S1B). Following air-puff, there was a significant increase in reactivity for both groups, with psilocin treated animals showing significantly greater PVN reactivity (2-way ANOVA: $F_{interaction}(20, 320) = 1.951$, $p = 0.0092$; $F_{treatment}(1,16) = 5.478$, $p = 0.0325$; Fig. 2C), with increases seen in both AUC ($t(16) = 2.798$, $p = 0.01$) and PP ($t(16) = 2.334$, $p = 0.03$) of the PVN reactivity trace compared to vehicle control (Fig. 2C). Additionally, psilocin treated males exhibited significantly lower velocity following air-puff as compared to vehicle control (2-way ANOVA: $F_{time}(9,189) = 10.12$, $p < 0.0001$; $F_{treatment}(1,21) = 6.246$, $p = 0.02$; Fig. 2D left) and less distance traveled compared to vehicle control (2-way ANOVA: $F_{time}(1,21) = 7.647$, $p = 0.01$; $F_{treatment}(1,21) = 4.896$, $p = 0.03$; Fig. 2D right).

Next, female PVN reactivity and threat-responding behavior were tested following the same timeline as above. At baseline, there were no differences seen between groups in locomotion or time spent in the center (Fig. S2A). Following the air puff, PVN reactivity was significantly increased, with no differences seen between groups (Fig. 3A). In the 10-seconds following the baseline air puff, animals showed increased velocity and distance traveled, but no differences were seen between groups (Fig. 3B). On the day of injection, psilocin treated females showed significantly less locomotion compared to vehicle group overall (2-way ANOVA: $F_{time}(14,266) = 39.80$, $p < 0.0001$; $F_{treatment}(1,19) = 8.943$, $p = 0.008$; Fig. S2B). For time spent in the center, changes across time were dependent on the treatment group (2-way ANOVA: $F_{interaction}(14,266) = 1.922$, $p = 0.024$; Fig. S2B). Following injection, PVN reactivity in the females was increased, however there were no differences between psilocin and vehicle groups (Fig. 3C). Similarly, all animals showed increased velocity and distance traveled in response to the air-puff, but no differences were seen between groups (Fig. 3D).

### Prolonged effects of psilocin on PVN reactivity and behavior

Given clinical findings suggesting prolonged effects of psychedelics following a single dose, male and female subjects were tested at 2- and 6/7-days post-injection to assess prolonged changes in behavior and PVN reactivity. In the males, at the 2-day follow up, there were no differences seen in locomotion or time spent in center (Fig. S1C). There was a significant increase in reactivity following the air-puff in both groups, with no differences seen between groups (Fig. 4A). Animals showed an increase in velocity within the 10-seconds following air-puff, however no differences were seen between groups in velocity or distance traveled (Fig. 4B). At the 6/7-day follow up, there were no differences in locomotion or time spent in center (Fig. S1D). There was a significant increase in PVN reactivity in response to the air-puff, with no differences observed between groups (Fig. 4C). Additionally, all animals showed an increase in velocity and distance traveled following air-puff, but no differences were seen between groups (Fig. 4D).

In the females, on the 2-day follow up, there were no differences seen between groups in locomotion (Fig. S2C). However, for time spent in center, there was an interaction of treatment and time with

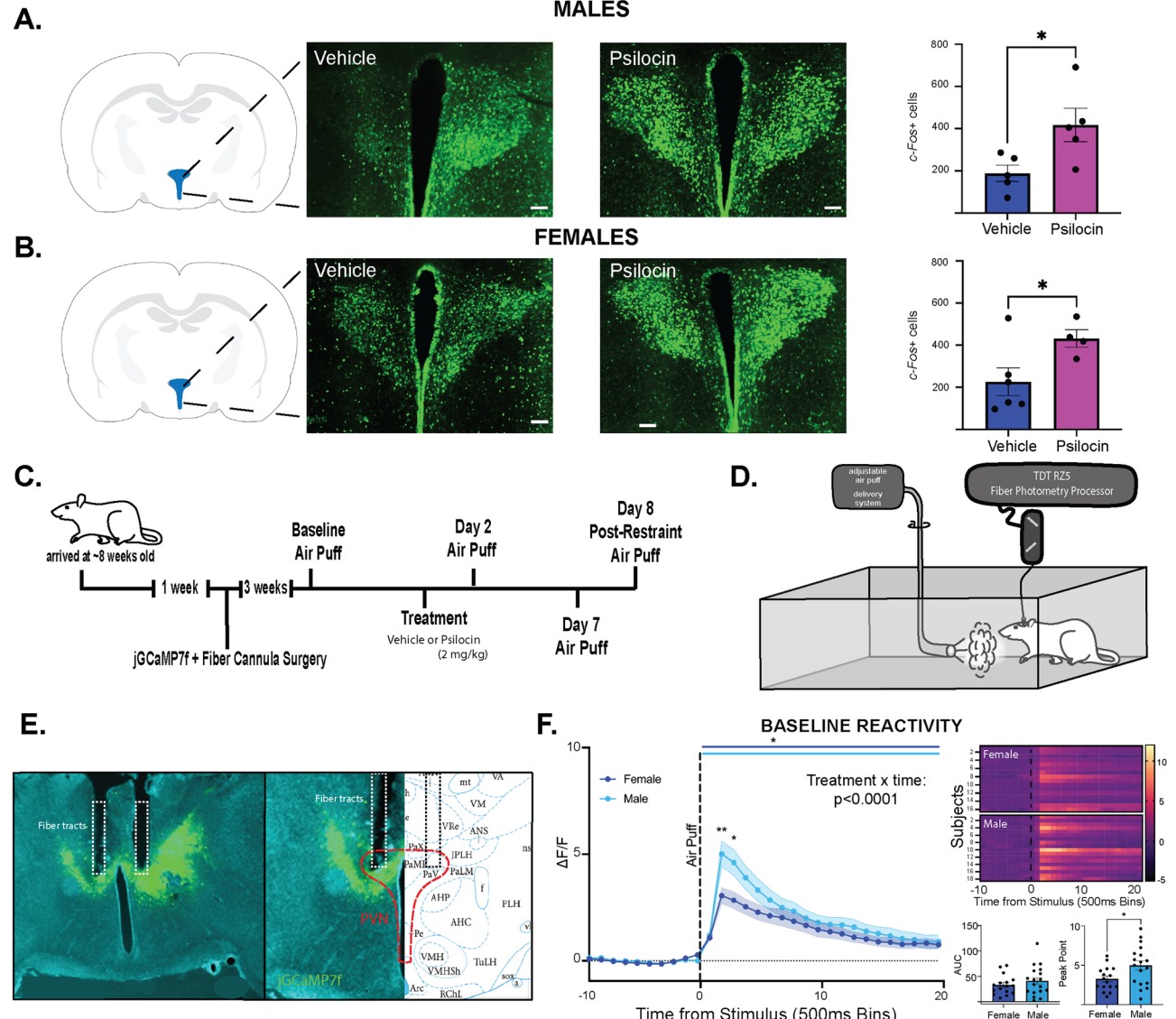

**Fig. 1 | Experimental details. A** Males: representative images showing c-Fos+ cells tagged with a green fluorescence protein (GFP) in the PVN. Scale bar = 100 μm. Each data point represents a single subject (average of c-Fos counts for 3 sections). Histograms display group averages +/- S.E.M. Comparisons were made using an unpaired 2-tailed t-test. Psilocin group $n = 5$; Vehicle group $n = 5$. **B** Females: representative images showing c-Fos+ cells tagged with a green fluorescence protein (GFP) in the PVN. Scale bar = 100 μm. Each data point represents a single subject (average of c-Fos counts for 3 sections). Histograms display group averages +/- S.E.M. Comparisons were made using an unpaired 2-tailed t-test. Psilocin group $n = 4$; Vehicle group $n = 6$. **C** Experimental timeline. Animals arrived at ~8 weeks old and were allowed to habituate for 1-week prior to surgeries. Bilateral injection of calcium sensor (jGCaMP7f-AAV9) and implantation of a dual tip fiber optic cannula were conducted. After allowing 3 weeks for viral transfection, animals underwent a series of fiber photometry experiments. **D** Illustration depicting the animal in open field box with air-puff apparatus and fiber photometry system. **E** Representative coronal section (left) and split cartoon schematic (right) showing fiber placement and GCaMP expression. **F** Baseline PVN reactivity Females ($n = 16$) vs. Males ($n = 18$): ΔF/F trace plots of changes in PVN fluorescence following exposure to a 500 ms air-puff. A 2-way ANOVA found a significant treatment x time interaction $p < 0.0001$. Šidák multiple comparisons found significant differences at time bin 12 ($p = 0.0038$) and 13 ($p = 0.0140$). Data points represent group averages within 500 ms binned window mean +/- S.E.M. (shaded area); Heatmaps (top right) comparing individual responses to air-puff (dotted line) in males and females. Average AUC and PP +/- S.E.M. (bottom right) compared by unpaired t-test. Each data point represents an individual subject. Comparisons were made using 2-tailed unpaired t-tests. *$p < 0.05$, **$p < 0.01$, ***$p < 0.001$, ****$p < 0.0001$. AUC = area under curve, PP = peak point, ΔF/F = change in fluorescence as a function of baseline fluorescence.

significant difference between vehicle and psilocin treated rats at minute 7 during habituation (2-way ANOVA: $F_{interaction}(14,266) = 2.208$, $p = 0.008$; Fig. S2C). At 2-days post-injection, there was a significant increase in PVN reactivity in response to air-puff, but no difference seen between groups (Fig. 5A). Velocity and distance traveled increased immediately following air-puff, but no differences were seen between groups (Fig. 5B). At the 6/7-day follow up, there were no differences between groups in locomotion or time spent in center (Fig. S2D). Following air-puff, there was a significant increase in PVN reactivity that did not differ between groups (Fig. 5C). Interestingly,

there was a significant interaction in velocity wherein psilocin-treated females showed greater velocity at 2 seconds compared to vehicle control (2-way ANOVA: $F_{interaction}(9,198) = 1.938$, $p = 0.04$; Fig. 5D), however, no differences were seen in distance traveled during that 10-second period (Fig. 5D).

**Effects of psilocin on PVN reactivity and threat responding following acute stress in male and females**

One potential appeal of psychedelic compounds is that they appear to provide prolonged therapeutic adaptations related to reductions in

# MALES

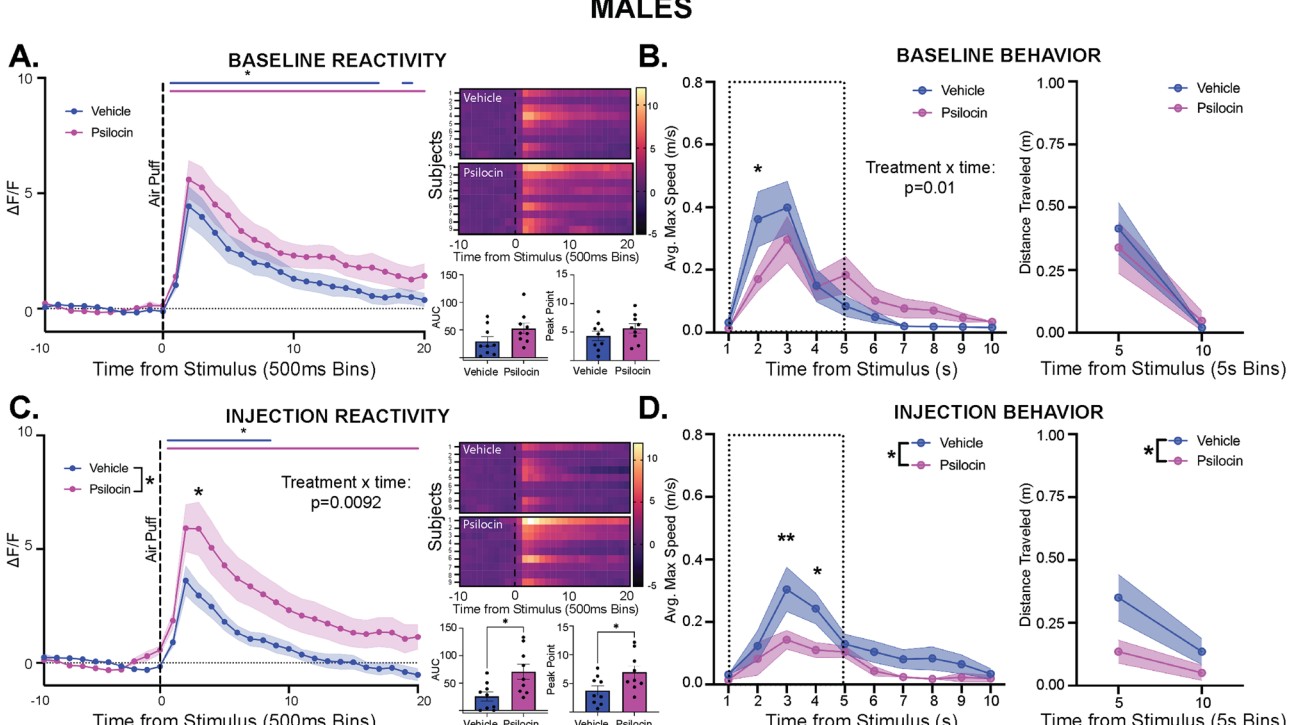

**Fig. 2 | Acute stimulus-evoked effects of psilocin on PVN reactivity and behavior in males.** Fiber photometry: Psilocin $n = 9$, Vehicle $n = 9$; Behavior: Psilocin $n = 11$, Vehicle $n = 12$; **A** Baseline PVN reactivity (left): ΔF/F trace plots of changes following 500 ms air-puff. A 2-way ANOVA was performed for statistical analysis. Data points represent 500 ms binned group averages with mean +/- S.E.M (shaded area).; Heatmaps (top right) comparing individual responses to air-puff (dotted line) in vehicle and psilocin groups.); Average AUC and PP + /- S.E.M. (bottom right) compared by 2-tailed unpaired t-tests between groups. Each data point represents an individual subject. **B** Average maximum speed (left) and distance traveled (right) after air-puff stimulus. A 2-way ANOVA was performed for statistical analysis. Data points are group averages +/- S.E.M. (shaded area) (**C**) Day of injection PVN reactivity (left): ΔF/F trace plots of changes following 500 ms air-puff. A 2-way ANOVA found a significant treatment x time interaction ($p = 0.0092$) and a main effect of treatment ($p = 0.0340$). Šidák multiple comparisons found significant differences

at time bin 13 ($p = 0.0149$). Data points represent group averages within 500 ms binned window +/- S.E.M. (shaded area); Heatmaps (top right) comparing individual responses to air-puff (dotted line) in vehicle and psilocin groups. Average AUC and PP + /- S.E.M. (bottom right) compared by 2-tailed unpaired t-test with a significant difference in AUC ($p = 0.0129$) and PP ($p = 0.0330$) between groups. Each data point represents an individual subject. **D** Average maximum speed (left) and distance traveled (right) after air-puff stimulus. A 2-way ANOVA found a main effect of treatment ($p = 0.0208$). Data points are group averages +/- S.E.M (shaded area). In each trace bin plot, a significant increase in ΔF/F was determined whenever the lower bound of the 99% CI was >0 with statistical significance shown as colored lines above each ΔF/F curve with colors corresponding to the respective binned traces. *$p < 0.05$, **$p < 0.01$, *** $p < 0.001$, **** $p < 0.0001$. AUC = area under curve, PP = peak point, ΔF/F = change in fluorescence as a function of baseline fluorescence, CI = confidence interval.

symptoms of anxiety and depression. Given that stress exposure can instigate differential activation of stress-responsive circuitry, we wanted to test whether prior psilocin administration altered PVN reactivity following exposure to an acute stressor, so animals underwent a 20-minute restraint prior to the air-puff photometry recording session. To confirm restraint-induced stress, animals were tail bled immediately following restraint to measure peripheral corticosterone (CORT) levels. Once blood was collected, animals were placed into the open field box for habituation prior to fiber photometry recordings to test the reactivity of PVN following an acute stress exposure. Following the 20-minute restraint stress CORT levels were elevated consistent with previous findings[31] in both males and females, however no significant effects of prior psilocin administration were found (Fig. 6A). In alignment with previous findings[32,33], females showed greater concentrations of CORT compared to males regardless of treatment (2-way ANOVA: $F_{sex}(1,34) = 51.52$, $p < 0.0001$; Fig. 6A).

In males, there were no differences seen in locomotion or time spent in the center across the entire recording session (Fig. S3A). Following acute restraint stress, the air-puff was able to elicit a significant increase in PVN reactivity, but no differences were seen between groups (Fig. 6B). Similarly, there was an acute increase in velocity and distance traveled following air-puff, but there were no differences between groups (Fig. 6C). In females, there were no differences seen between groups in locomotion or time spent in the

center across the session (Fig. S3B). Following restraint stress, there was a significant increase in PVN reactivity, however, no differences were seen between groups (Fig. 6D). Similarly, there was an acute increase in velocity and distance traveled following air-puff, but there were no differences between groups (Fig. 6E).

## Active vs. passive threat responding in males and females
Upon exposure to the air-puff stimulus, animals either employed an active or passive threat response (Fig. 7A). An active response was defined as darting behavior involving an escape from the quadrant within which the air-puff was administered. A passive response was defined as immobility that did not involve an active escape from the quadrant where the air puff occurred. In the males, 58.3% of the vehicle group and 44.4% of the psilocin group employed an active response to the air puff at baseline (Fig. 7B) with a significantly higher proportion of vehicle active responders at baseline ($p = 0.01$) and injection ($p < 0.0001$, Fig. 7B). In the females, 66.7% of the vehicle group and 83.3% of the psilocin group employed an active response at baseline (Fig. 7C), with a significantly higher proportion of psilocin active responders at baseline ($p = 0.01$) and 6/7-days post injection ($p < 0.0001$), and greater proportion of vehicle active responders on the day of injection ($p < 0.0001$) and 2-days post-injection ($p = 0.03$, Fig. 7C). In the females, most of the animals in the psilocin group employed an active threat responding strategy, and therefore, groups

# FEMALES

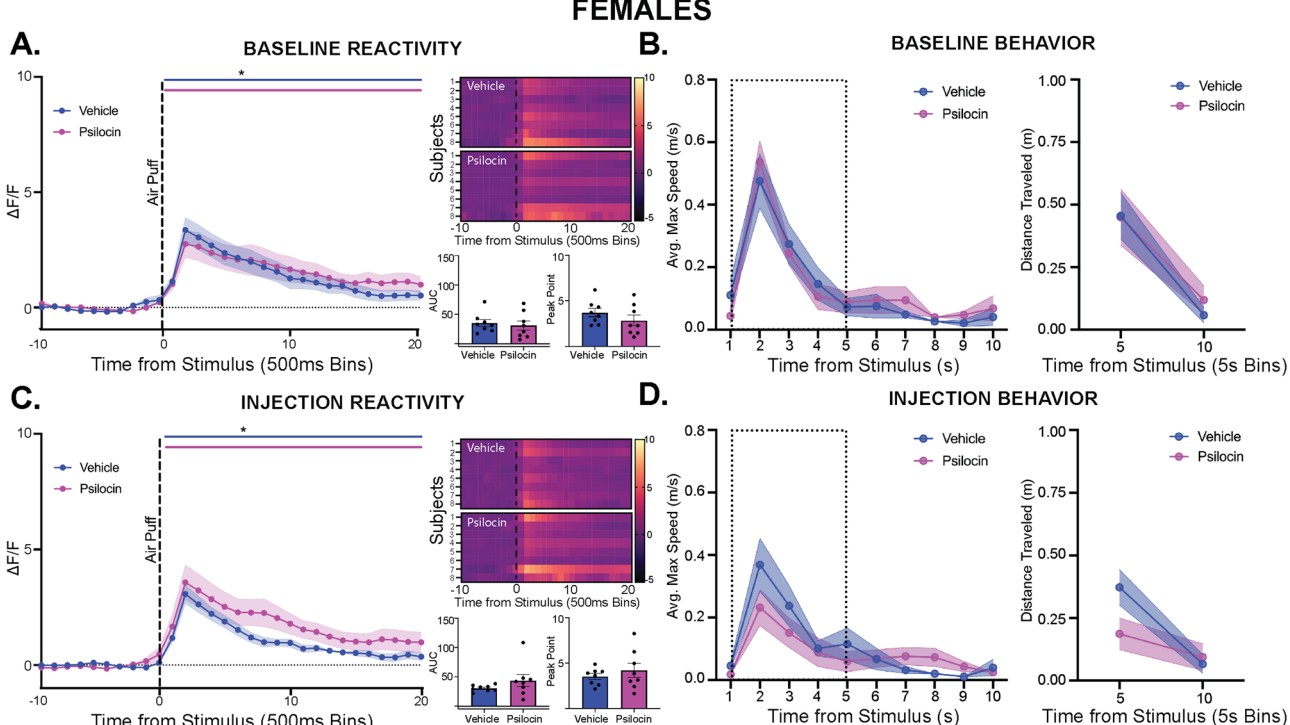

**Fig. 3 | Acute stimulus-evoked effects of psilocin on PVN reactivity and behavior in females.** Fiber photometry: Psilocin $n = 8$, Vehicle $n = 8$; Behavior: Psilocin $n = 12$, Vehicle $n = 12$; **A** Baseline PVN reactivity (left): ΔF/F trace plots of changes following 500 ms air-puff. Comparisons were made using a 2-way ANOVA. Data points represent 500 ms binned group averages with mean +/- S.E.M. (shaded area); Heatmaps (top right) comparing individual responses to air-puff (dotted line) in vehicle and psilocin groups. Average AUC and PP +/- S.E.M. (bottom right) compared by 2-tailed unpaired t-tests between groups. Each data point represents an individual subject. **B** Average maximum speed (left) and distance traveled (right) after air-puff stimulus. A 2-way ANOVA was performed for statistical analysis. Data points are group averages +/- S.E.M. (shaded area) (**C**) Day of injection PVN reactivity (left): ΔF/F trace plots of changes following exposure to 500 ms air-puff. A 2-way ANOVA was performed for statistical analysis. Data points represent group averages within 500 ms binned window +/- S.E.M. (shaded area); Heatmaps (top right) comparing individual responses to air-puff (dotted line) in vehicle and psilocin groups.; Average AUC and PP +/- S.E.M (bottom right) compared by 2-tailed unpaired t-tests between groups. Each data point represents an individual subject. **D** Average maximum speed (left) and distance traveled (right) after air-puff stimulus. Comparisons were made using a 2-way ANOVA. Data points are group averages +/- S.E.M (shaded area). In each trace bin plot, a significant increase in ΔF/F was determined whenever the lower bound of the 99% CI was >0 with statistical significance shown as colored lines above each ΔF/F curve with colors corresponding to the respective binned traces. *$p < 0.05$, **$p < 0.01$, ***$p < 0.001$, ****$p < 0.0001$. AUC = area under curve, PP = peak point, ΔF/F = change in fluorescence as a function of baseline fluorescence, CI = confidence interval.

could not be further split into active and passive responding subgroups without compromising power. However, the proportion of active vs. passive responders was more evenly split between the males, allowing for secondary analysis to determine the effects of psilocin within these threat-responding subgroups.

## Psilocin produces increases in PVN reactivity in active responding but not passive responding males

To further assess the active and passive responses to threat phenotype, additional analyses were conducted within each group across the entire behavior period, including the 10 min habituation period and 5 min post puff ISI. Here, we examined the effects on immobility and time spent in the center as these behaviors are thought to be involved in the effect. In the active responding males, there were no differences between groups in time spent in the center or time spent immobile at baseline (Fig. S4A). Following air-puff, there were significant increases in PVN reactivity, but no differences seen between treatment groups (Fig. 8A). All animals displayed an increase in velocity and distance traveled following the air-puff, however there were no differences between groups (Fig. 8B). On the day of injection, there were no differences between groups in time spent in center. However, the active responding psilocin-treated males showed greater immobility compared to vehicle control (2-way ANOVA: $F_{interaction}(14, 84) = 2.382$, $p = 0.007$; $F_{time}(14, 84) = 6.672$, $p < 0.0001$; $F_{treatment}(1, 6) = 10.19$, $p = 0.01$, Fig. S4B). Following air-puff administration, the active

responding psilocin treated animals showed a significantly larger response in PVN reactivity (2-way ANOVA: $F_{interaction}(29, 203) = 8.247$, $p < 0.0001$; $F_{treatment}(1, 7) = 12.80$, $p = 0.009$, Fig. 6C), with a significant increase in AUC ($t(7) = 2.992$, $p = 0.02$, Fig. 8C) compared to vehicle control. Interestingly, psilocin active responders showed a significantly greater reduction in active threat responding behavior (increased passive responding) compared to vehicle control (Fisher's: $p = 0.0004$, Fig. 8C inset), suggesting that increased PVN reactivity was associated with decreases in active threat responding behavior. All animals displayed an increase in velocity and distance traveled following the airpuff, and there were no differences between groups (Fig. 8D). No differences were seen between vehicle and psilocin active responding males at the 2- and 6/7-day follow-up.

In the passive-responding males, there were no differences seen in immobility time or time spent in the center at baseline (Fig. S4C). At baseline, there were significant increases in PVN reactivity to the air puff seen in both groups, with a greater AUC but not PP observed in the psilocin treated animals as compared to vehicle control ($t(7) = 2.410$, $p = 0.046$; Fig. 9A). All psilocin treated passive responders showed a subtle increase in velocity and distance traveled following air-puff, but there were no differences between groups seen (Fig. 9B). On the day of injection, there was a significant interaction of time and treatment for time spent in center seen in the psilocin treated males (2-way ANOVA: $F_{interaction}(14, 112) = 2.540$, $p = 0.003$, Fig. S4D). Additionally, psilocin treated passive males spent significantly more time immobile than

# MALES

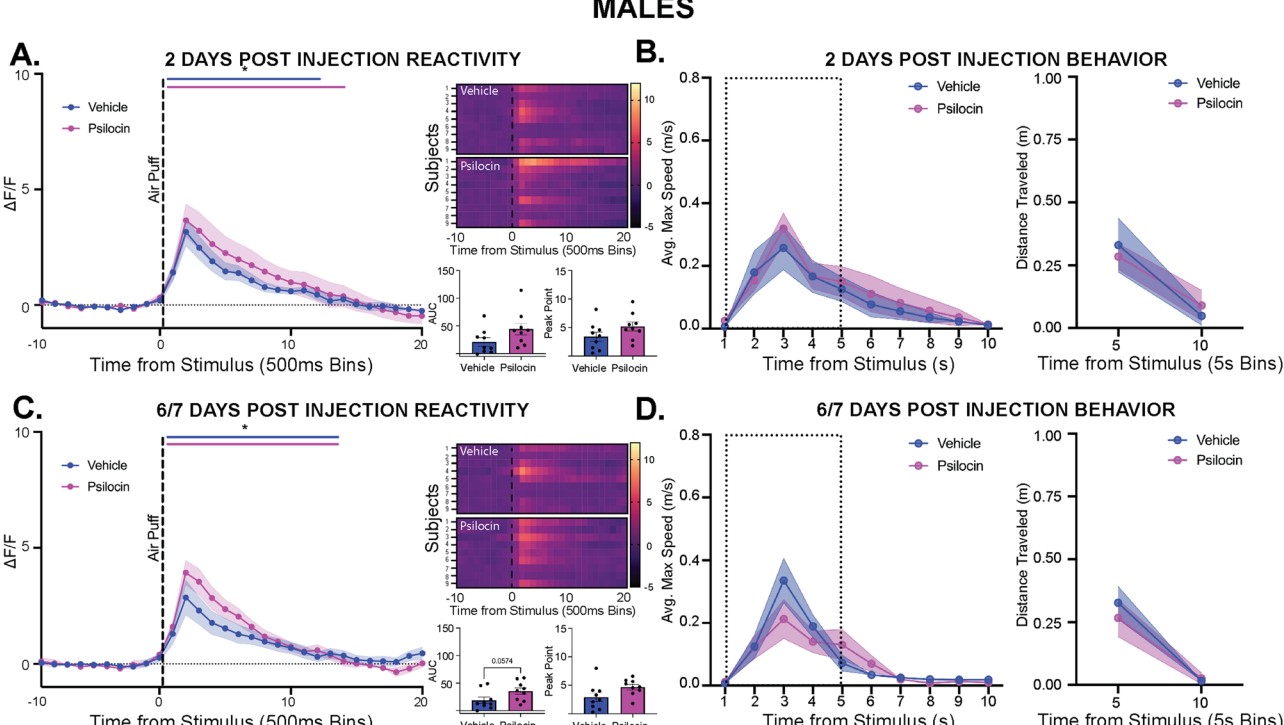

**Fig. 4 | Prolonged effects of psilocin on PVN reactivity and behavior in males.**
Fiber photometry: Psilocin $n = 9$, Vehicle $n = 9$; <u>Behavior</u>: Psilocin $n = 11$, Vehicle $n = 12$; **A** 2-days post-injection PVN reactivity (left): ΔF/F trace plots of changes following exposure to a 500 ms air-puff. A 2-way ANOVA was performed for statistical analysis. Data points represent group averages within 500 ms binned window +/- S.E.M. (shaded area); Heatmaps (top right) comparing individual responses to air-puff stimulus (dotted line) in vehicle and psilocin groups. Average AUC and PP +/- S.E.M. (bottom right) compared by 2-tailed unpaired t-tests between groups. Each data point represents an individual subject. **B** Average maximum speed (left) and distance traveled (right) after air-puff stimulus. A 2-way ANOVA was performed for statistical analysis. Data points are group averages +/- S.E.M. (shaded area) (**C**) 6/7-days post-injection PVN reactivity (left): ΔF/F trace plots of changes following exposure to a 500 ms air-puff. A 2-way ANOVA was performed for statistical

analysis. Data points represent group averages within 500 ms binned window +/- S.E.M. (shaded area); Heatmaps (top right) comparing individual responses to air-puff stimulus (dotted line) in vehicle and psilocin groups. Average AUC and PP +/- S.E.M. (bottom right) compared by 2-tailed unpaired t-tests between groups. Each data point represents an individual subject. **D** Average maximum speed (left) and distance traveled (right) after air-puff stimulus. A 2-way ANOVA was performed for statistical analysis. Data points are group averages +/- S.E.M. (shaded area). In each trace bin plot, a significant increase in ΔF/F was determined whenever the lower bound of the 99% CI was >0 with statistical significance shown as colored lines above each ΔF/F curve with colors corresponding to the respective binned traces *$p < 0.05$, **$p < 0.01$, ***$p < 0.001$, ****$p < 0.0001$. AUC = area under curve, PP = peak point, ΔF/F = change in fluorescence as a function of baseline fluorescence, CI = confidence interval.

---

vehicle control (2-way ANOVA: $F_{time}(14, 112) = 12.37$, $p < 0.0001$; $F_{treatment}(1, 8) = 11.14$, $p = 0.01$, Fig. S4D). Following air-puff administration, there was a significant increase in reactivity following air-puff in both groups, with no differences between vehicle and psilocin treated animals (Fig. 9C). Interestingly, in the vehicle treated passive males there was a greater reduction in passive threat responding (i.e., increased darting) in response to the air-puff compared to baseline, with no changes in threat responding seen in psilocin passive responders (Fisher's: $p < 0.0001$, Fig. 9C inset). Additionally, in the vehicle treated passive responder males there was an acute increase in velocity and distance traveled following air-puff, but there were no differences between groups (Fig. 9D). These findings suggest that drug sensitivity is potentially driven by male subjects that employed an active baseline threat response, however, as behavioral subgroups were identified after data collection, sample sizes were out of our control and are smaller than other experimental groups.

## Discussion

In the current study, we utilized immunohistochemistry and fiber photometry in combination with in-vivo behavioral recording to determine the effects of the psychedelic compound, psilocin, on stimulus-independent activity and stimulus-evoked reactivity within the paraventricular nucleus of the hypothalamus (PVN). This study provides insight involving the real-time analysis of changes in threat-

responding behavior paired with simultaneous fluctuations in calcium dynamics within the PVN after administration of the serotonergic psychedelic psilocin. This work provides a deeper understanding of generalized changes in PVN reactivity following psychedelic administration and a valuable foundation upon which future studies will develop causal inferences regarding the role of changes in PVN reactivity in the behavioral effects of psychedelics. However, one limitation of this study is that there were no cell-type specific approaches employed, and therefore specific mechanisms underlying changes in PVN reactivity remain to be determined. In both males and females, we found that psilocin administration (2 mg/kg, s.c.) produced increases in stimulus-independent activity as measured by c-Fos expression within the PVN. Next, utilizing fiber photometry we found that the air-puff stimulus evoked an increase in PVN reactivity at baseline for both males and females, with a greater level of activation seen in males as compared to females. Following injection, psilocin-treated males showed increased PVN reactivity compared to vehicle control. Interestingly, the increases in PVN reactivity seen in males corresponded to decreases in velocity and distance traveled within the same 10-second timeframe. In females, there were no differences between the vehicle and psilocin groups in PVN reactivity, and, additionally, no differences in velocity or distance traveled immediately following air-puff. While this observed sex-specificity is in contrast with the c-Fos data, it is important to note the differences between stimulus-independent

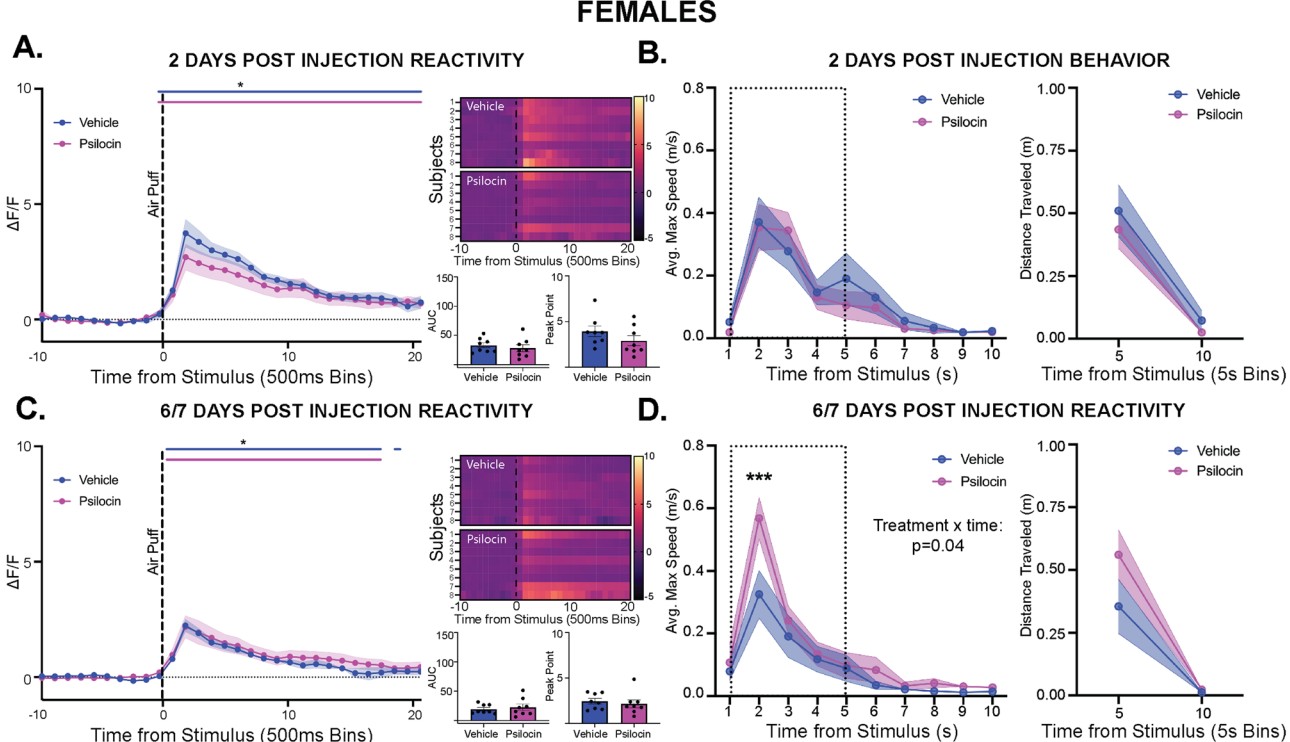

**Fig. 5 | Prolonged effects of psilocin on PVN reactivity and behavior in females.** Fiber photometry: Psilocin $n = 8$, Vehicle $n = 8$; Behavior: Psilocin $n = 12$, Vehicle $n = 12$; **A** 2-days post-injection PVN reactivity (left): ΔF/F trace plots of changes following exposure to a 500 ms air-puff. A 2-way ANOVA was performed for statistical analysis. Data points represent group averages within 500 ms binned window +/- S.E.M. (shaded area); Heatmaps (top right) comparing individual responses to air-puff stimulus in vehicle and psilocin groups. Average AUC and PP +/- S.E.M. (bottom right) compared by unpaired 2-tailed t-tests between groups. Each data point represents an individual subject. **B** Average maximum speed (left) and distance traveled (right) after air-puff stimulus. A 2-way ANOVA was performed for statistical analysis. Data points are group averages +/- S.E.M. (shaded area) (**C**) 6/7-days post-injection PVN reactivity (left): ΔF/F trace plots of changes following exposure to a 500 ms air-puff. A 2-way ANOVA was performed for statistical analysis. Data points represent group averages within 500 ms binned window +/- S.E.M. (shaded area); Heatmaps (top right) comparing individual responses to air-puff stimulus (dotted line) in vehicle and psilocin groups. H Average AUC and PP +/- S.E.M. (bottom right) compared by unpaired 2-tailed t-tests between groups. Each data point represents an individual subject. **D** Average maximum speed (left) and distance traveled (right) after air-puff stimulus. A 2-way ANOVA found an interaction between treatment and time ($p = 0.0486$). Šidák multiple comparisons revealed a significant difference at 2 seconds ($p = 0.0001$). Data points are group averages +/- S.E.M (shaded area). In each trace bin plot, a significant increase in ΔF/F was determined whenever the lower bound of the 99% CI was >0 with statistical significance shown as colored lines above each ΔF/F curve with colors corresponding to the respective binned traces *$p < 0.05$, **$p < 0.01$, ***$p < 0.001$, ****$p < 0.0001$. AUC = area under curve, PP = peak point, ΔF/F = change in fluorescence as a function of baseline fluorescence, CI = confidence interval.

activation and stimulus-induced activation. Stimulus-induced changes can be impacted by a variety of different mechanisms and could point towards differential cell-type expression or innervating circuitry between sexes that could govern differential stimulus-induced activation. Future work will investigate upstream mediators of the PVN as well as employ a more cell-type specific approach to address these questions.

Given the observed persistent effects of psychedelics in clinical trials and previously published work[30], we assessed PVN reactivity at multiple time points following injection. In males, acutely elevated PVN reactivity to the air puff following injection was restored at the 2- and 7-day follow-up time points. In both males and females, PVN reactivity remained consistent between groups at 2- and 7 days following injection. To assess the effects of prior psilocin exposure on PVN reactivity following an acute stressor, we utilized a 20-minute restraint stress immediately followed by fiber photometry recording in the PVN. We found that prior psilocin exposure did not alter PVN reactivity following acute restraint stress. To ensure that 20-minute restraint was sufficient to elicit a stress response, serum analyses were performed demonstrating levels of CORT indicative of a stressful experience[31] in both males and females following restraint, with higher levels seen in females. Prior exposure to psilocin did not alter serum stress-associated CORT levels in males or females. These data suggest that

psilocin does not interfere with the hormonal response to acute stress or the ability of the PVN to increase reactivity following acute stress exposure.

The PVN appears to be a highly relevant region as it pertains to the subcortical effects of psychedelic drug compounds. Previous work has shown that 5-HT$_{2AR}$ agonizts such as DOI, produce increases in c-Fos expression within the PVN[34] and increased plasma levels of corticosterone[35]. Similarly, studies have shown that administration of the 5-HT$_{2AR}$ agonist psychedelic lysergic acid diethylamide (LSD) produced substantial increases in c-Fos expression in the PVN, with increases seen up to 4 hr following administration, the latest time point examined in the study[36]. Clinical work suggests that dysregulation of the PVN is a characteristic of several psychiatric disorders. For instance, post-mortem studies have shown increased CRF mRNA levels in the PVN of patients with MDD[37]. Patients with major depressive disorder (MDD) have been shown to have increased CRF+ neurons in the PVN[38]. Similarly, it has been shown that patients with MDD and bipolar disorder exhibit increases in oxytocin, and vasopressin neurons[39]. Additionally, patients with MDD and BD show selective neuronal loss in the PVN[40]. Preclinical studies have shown that exposure to a stressor promotes increases in oxytocin (OXT) release both within the PVN and systemically[41]. Additionally, chronic variable stress has been shown to double synapses onto CRF neurons within the PVN

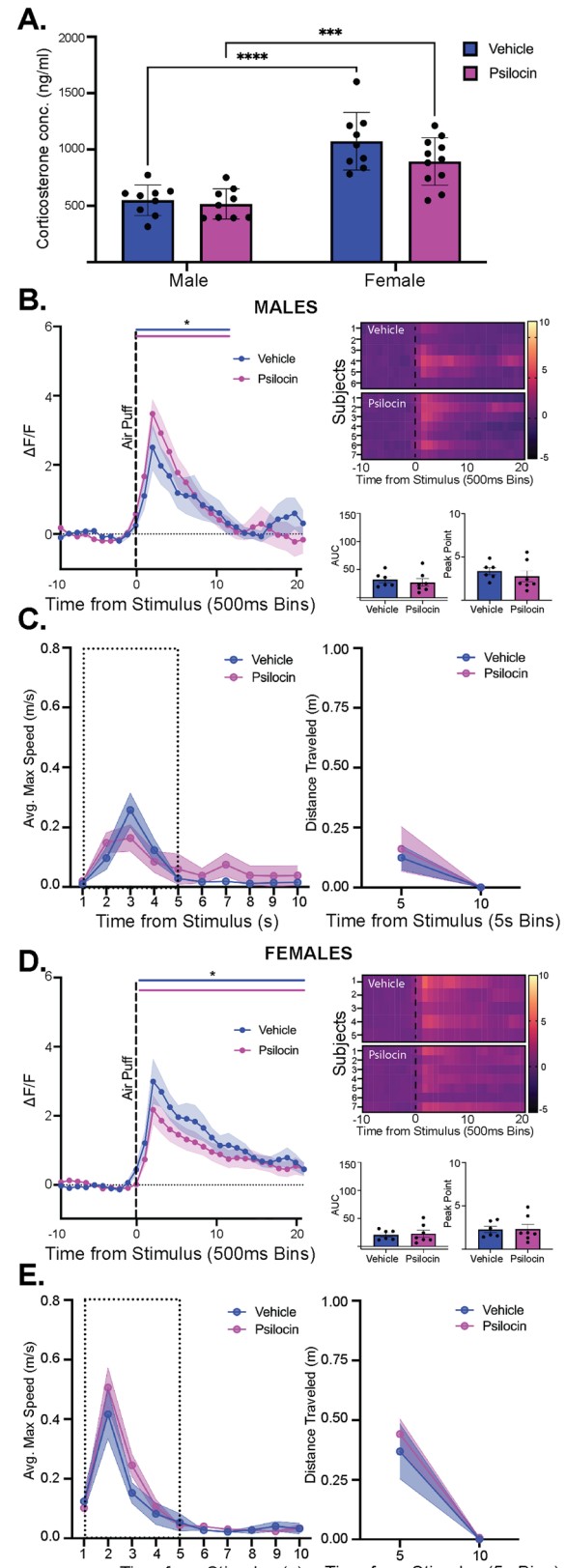

**Fig. 6 | Effects of psilocin on PVN reactivity following an acute stress in male and females. A** Average male (Psilocin $n = 9$; Vehicle n = 9) and female (Psilocin $n = 11$, Vehicle $n = 9$) plasma corticosterone levels. 2-way ANOVA revealed a main effect of sex ($p < 0.0001$). Each data point represents an individual subject. Mean +/- S.E.M. compared between groups. **B** Male post-restraint PVN reactivity (left): ΔF/F trace plots of changes following exposure to a 500 ms air-puff A 2-way ANOVA was performed for statistical analysis. Data points represent group averages within 500 ms binned window +/- S.E.M. (shaded area); Heatmaps (top right) comparing individual responses to air-puff stimulus (dotted line) in vehicle and psilocin groups. Average AUC and PP +/- S.E.M. (bottom right) compared by unpaired 2-tailed t-test between groups. Each data point represents an individual subject (Psilocin n = 7, Vehicle $n = 6$). **C** Male: Average maximum speed (left) and distance traveled (right) following air-puff stimulus. 2-way ANOVA was performed for statistical analysis. Data points are group averages +/- S.E.M. (shaded area) (Psilocin $n = 9$, Vehicle $n = 9$) (**D**) Female post-restraint PVN reactivity (left): ΔF/F trace plots of changes following exposure to a 500 ms air-puff. 2-way ANOVA was performed for statistical analysis. Data points represent group averages within 500 ms binned window +/- S.E.M. (shaded area); Heatmaps (top right) comparing individual responses to air-puff stimulus (dotted line) in vehicle and psilocin groups. Average AUC and PP +/- S.E.M. (bottom right) compared by unpaired 2-tailed t-test between groups. Each data point represents an individual subject. (Psilocin $n = 7$, Vehicle $n = 5$). **E** Female: Average maximum speed (left) and distance traveled (right) after air-puff stimulus. 2-way ANOVA was performed for statistical analysis. Data points are group averages +/- S.E.M. (shaded area) (Psilocin $n = 11$, Vehicle n = 9). In each trace bin plot, a significant increase in ΔF/F was determined whenever the lower bound of the 99% CI was >0 with statistical significance shown as colored lines above each ΔF/F curve with colors corresponding to the respective binned traces *$p < 0.05$, **$p < 0.01$, ***$p < 0.001$, ****$p < 0.0001$. AUC = area under curve, PP = peak point, ΔF/F = change in fluorescence as a function of baseline fluorescence, CI = confidence interval.

in rats[42]. Given the prominent role of the PVN in HPA axis activation, known activation of PVN following administration of 5-HT$_{2AR}$ agonist psychedelics, along with the well-documented dysregulation of the HPA axis associated with psychiatric illness, the PVN is a central site that likely contributes to the clinically observed therapeutic effects of psychedelic compounds.

The current study provides insight into in-vivo PVN reactivity to an aversive stimulus and identifies a sex-specific mechanism of action for psilocin administration to alter PVN calcium dynamics and changes in threat responding immediately following exposure to the stimulus. Here, we showed that psilocin administration produced increases in reactivity within the PVN in males, but no significant effects were seen in females, and no differences in either sex were seen at more prolonged time points. Interestingly, these effects are in contrast to previous findings from our group, wherein we found acute sex-specific effects of psilocin administration causing increased CeA reactivity in females only[30]. One potential mechanism for this could involve known inhibitory PVN→CeA circuitry, wherein oxytocinergic projections from the PVN to the CeA activate GABAergic cells in the lateral CeA (CeL) that then inhibit output neurons of the CeA located in the medial CeA (CeM)[43]. It is possible that psilocin produces sex-specific activation of this circuit in males such that an increase in acute reactivity in PVN→CeA circuitry that is having an inhibitory effect on CeM neurons would reduce/prevent an increased activation in the CeA in males.

Another potential circuit mediating these effects is a disynaptic hippocampus→bed nucleus of the stria terminalis (BNST)→PVN projection that has been shown to have an inhibitory effect on the PVN[44]. One study showed that following exposure to a stressor, there is an increase in CA1 spine density in males and a decrease in females[45]. Given that CA1 pyramidal neurons have been shown to display high levels of 5-HT$_{2AR}$ expression, and activation of these 5-HT$_{2A}$ receptors increases the firing of these neurons[46], differences in density could contribute to differential activation of the known inhibitory hippocampus→BNST→PVN circuit. However, the lack of cell-type specificity is a limitation of the current study, and further work is needed to fully characterize the cell-type specific mechanisms behind the increases in PVN reactivity seen in response to the air puff. Additionally, circuit mapping to identify potential upstream mediators of the observed effects will be an important future direction for this work.

Previous work in the lab showed that a single administration of psilocin produced prolonged decreases in central amygdala reactivity

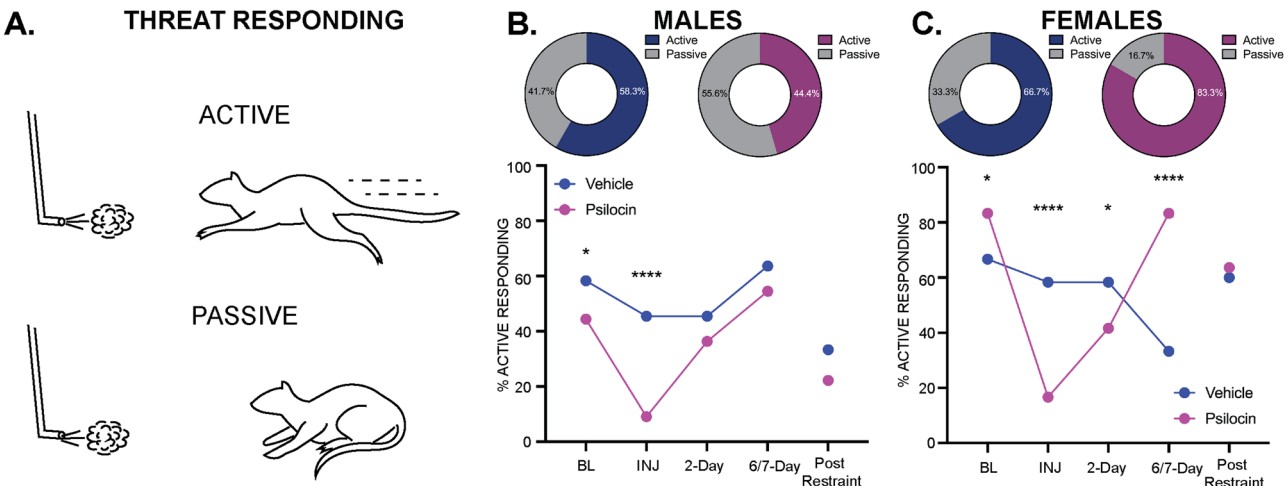

**Fig. 7 | Active vs. passive threat responding in males and females. A** Illustration depicting active vs. passive responding to the air-puff stimulus. **B** Males (Psilocin *n* = 11; Vehicle *n* = 12): pie chart (top) showing the proportion of active vs. passive responders at baseline. Line graph (bottom) comparing changes in the proportion of active responders in male psilocin and vehicle groups across each recording session. Each data point represents the proportion of active responders at each time period. Statistical analysis comparisons utilized a 2-sided Fisher's exact test found significant differences between groups at baseline ($p = 0.0127$) and injection ($p < 0.0001$). **C** Females (Psilocin *n* = 12; Vehicle *n* = 12): pie chart (top) showing the proportion of active vs. passive responders at baseline. Line graph (bottom) comparing changes in the proportion of active responders in male psilocin and vehicle groups across each recording session. Each data point represents the proportion of active responders at each time period. Statistical analysis comparisons using a 2-sided Fisher's exact test found significant differences between groups at baseline ($p = 0.0138$), injection ($p < 0.0001$), 2-day ($p = 0.0336$), and 7-day ($p < 0.0001$). *$p < 0.05$, **$p < 0.01$, ***$p < 0.001$, ****$p < 0.0001$. BL = baseline, INJ = injection.

in male, but not female Sprague Dawley rats[30]. In the present study, no such effects were found, suggesting that acute stress-induced activation remained unperturbed. Although we did not find any effect of psilocin on prolonged PVN reactivity, these data have been included as we find the lack of prolonged effects within the PVN to be of interest. These findings demonstrate stability in PVN reactivity to an aversive stimulus that is maintained following psilocin administration. The PVN is an essential region mediating autonomic functioning and orchestrating sympathetic nervous system activation. Previous work has suggested that PVN functioning is influenced by complex local excitatory and inhibitory circuitry[47,48]. For instance, it has been shown that local nitric oxide and y-aminobutyric acid (GABA) containing neurons within the PVN function to promote tonic inhibition over sympathetic outflow. This regulation of neuronal activity within the PVN is crucial to avoid hypertension and other, potentially lethal cardiovascular outcomes[49–52]. Thus, given the tightly regulated activity within the PVN due it's heavy involvement in autonomic and cardiovascular functioning, it is possible that a single administration of a drug would not produce any enduring changes in function. This may also explain why exposure to the 20-minute restraint stress didn't alter PVN reactivity to the air-puff stimulus. Stress exposure promotes excitation of corticotropin-releasing factor (CRF) expressing neurons within the PVN, leading to HPA-axis activation. Recent work suggests that within the PVN there are also CRF sensitive neurons expressing the corticotropin releasing factor receptor 1 (CRF$_{R1}$) that serve as an inhibitory feedback mechanism within the PVN to prevent hyperreactivity of the PVN and as a result the HPA axis[48]. Given the observed increases in peripheral corticosterone following restraint stress, it is likely that PVN CRF release would also have activated these CRF$_{R1}$ receptors in an attempt to dampen hyperreactivity. This could also account for the reactivity of the PVN not being altered following the acute restraint stress. However, it is possible that the observed stability in PVN reactivity may not be seen in subjects modeling psychiatric disease states, for instance, following acute stress exposure. Future studies will examine the effects of psilocin on PVN reactivity within subjects demonstrating anxiogenic pathology to better understand how these drug effects are modulated by psychiatric disease states.

Here, we replicated previous findings that females primarily employ an active threat response[30,53,54], whereas males are more evenly split between active and passive responding. Interestingly, it appears that active responding males showed slightly greater basal PVN reactivity than the passive responders, with a similar trend seen in the max speed following air-puff. This discrepancy between active and passive responder PVN reactivity may provide clues for future work investigating the neurobiological mechanisms underlying these divergent behavioral phenotypes. Here, we showed that active responding psilocin-treated males exhibited a significant increase in PVN reactivity compared to active vehicle, while passive responders showed no difference between groups, suggesting that increases in PVN reactivity seen in males were driven by the males that exhibited active responding at baseline. These findings replicate previous findings showing an effect of psilocin on brain region reactivity in active, but not passive responding males[30]. Together, these findings demonstrate that an animal's implicit stress-responding strategy and the underlying neurobiology driving this behavioral adaptation may be predictive of drug sensitivity, which could have implications for the clinical use of psilocybin and other psychedelic compounds. While these findings suggest that threat responding strategy may be predictive of psychedelic drug sensitivity, a limitation of the study is that subgroups were determined post hoc and were therefore outside of our control. As a result, sample sizes were relatively small. Given these findings, understanding the neurobiological underpinnings governing this active threat response may offer clues to the mechanisms behind psilocybin's effects. Recent work suggests there is divergent circuitry from the paraventricular nucleus of the thalamus (PVT) to the CeA and nucleus accumbens (NAc) controlling active vs. passive responding to a stressor, specifically PVT→CeA circuitry promoting passive responding and PVT→NAc circuitry promoting active responding[55]. It has also been shown that females have a greater spine density and proportion of large spines within the NAc[56,57]. Together these findings point to a potential explanation for the increased prevalence of active responding in females. Activation of the PVT→NAc pathway along with a NAc that includes increased structural and functional susceptibility to engagement could promote active vs. passive responding in females.

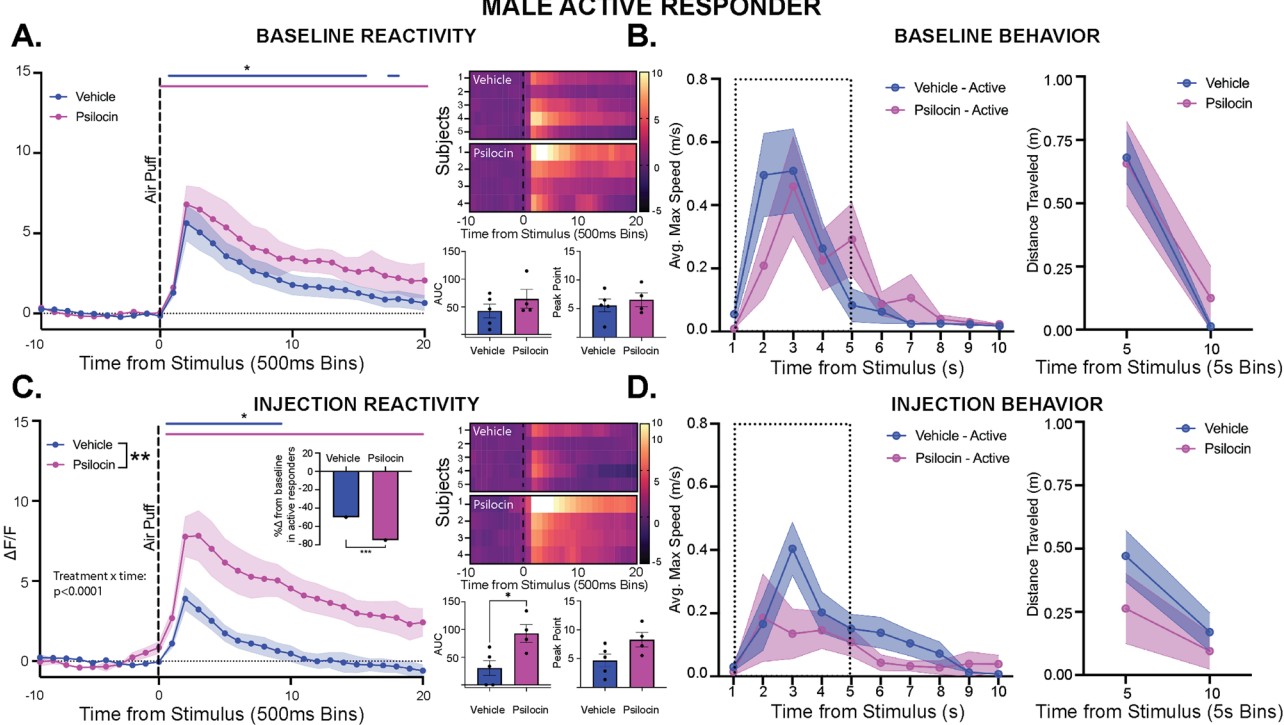

**Fig. 8 | Psilocin produces increases in PVN reactivity in active responding males.** Fiber photometry: Psilocin active $n = 4$, Vehicle active $n = 5$, Psilocin passive $n = 5$, Vehicle passive $n = 4$; Behavior: Psilocin active $n = 4$, Vehicle active $n = 6$, Psilocin passive $n = 7$, Vehicle passive $n = 6$; (**A**) Baseline PVN reactivity (left): ΔF/F trace plots of changes following exposure to a 500 ms air-puff. 2-way ANOVA was performed for statistical analysis. Data points represent group averages within 500 ms binned window +/- S.E.M. (shaded area); Heatmaps (top right) comparing individual responses to air-puff stimulus (dotted line) in vehicle and psilocin groups. Average AUC and PP +/- S.E.M. (bottom right) compared by unpaired 2-tailed t-test between groups. Each data point represents an individual subject. **B** Average maximum speed (left) and distance traveled (right) following air-puff stimulus. 2-way ANOVA was performed for statistical analysis. Data points are group averages +/- S.E.M. (shaded area) (**C**) PVN reactivity (left): ΔF/F trace plots of changes following exposure to a 500 ms air-puff. 2-way ANOVA significant treatment x time interaction ($p < 0.0001$) and main effect of treatment ($p = 0.0091$). Data points represent group averages within 500 ms binned window +/- S.E.M. (shaded area); Heatmaps (top right) comparing individual responses to air-puff stimulus (dotted line) in vehicle and psilocin groups. Average AUC and PP +/- S.E.M (bottom right) compared by unpaired 2-tailed t-test between groups, AUC ($p = 0.0202$). Each data point represents an individual subject. Inset: Histogram reflecting change in the proportion of active responding animals compared to baseline. 2-sided Fisher's exact test revealed a significant difference between groups ($p = 0.0004$). **D** Average maximum speed (left) and distance traveled (right) following air-puff stimulus. 2-way ANOVA was performed for statistical analysis. Data points are group averages + / - S.E.M. (shaded area). In each trace bin plot, a significant increase in ΔF/F was determined whenever the lower bound of the 99% CI was >0 with statistical significance shown above each ΔF/F curve with colors corresponding to the respective binned traces *$p < 0.05$, **$p < 0.01$, ***$p < 0.001$, ****$p < 0.0001$. AUC = area under curve, PP = peak point, ΔF/F = change in fluorescence as a function of baseline fluorescence, CI = confidence interval.

Interestingly, we showed in this study that males exposed to psilocin exhibited a significant reduction in active responding velocity behavior. Activation of CRF+ neurons within the PVN has been implicated in peripheral stress response. CRF+ neurons activate CRF receptors leading to peripheral sympathetic nervous system activation[58,59]. As activation of the PVN increased during the 10-second window in which rats displayed decreased max speed and active/darting behavior, it is possible that the psilocin induced increases in activation we found are not targeting CRF neurons within the PVN. One potential mechanism behind an increased activation of PVN neurons resulting in decreased flight behavior following psychedelic drug administration could involve oxytocin (OT) neurons within the PVN. Previous studies have suggested that OT administration increases serotonin release in the dorsal raphe (DR) promoting a reduction in anxiety-like behavior, an effect that was blocked with a 5-HT$_{2A/2C}$ antagonist[60]. This work suggests that these 5-HT$_{2A}$R expressing neurons within the DR may also express oxytocin receptors, so presumably, the same mechanisms may occur in 5-HT$_{2A}$R expressing neurons in the PVN. Additionally, it has been shown that the endogenous release of OT within the PVN promotes inhibition of CRF+ cells[61]. Given that psilocin is a potent 5-HT$_{2A}$R agonist, activation of these neurons could also activate local OT inhibitory circuitry which could explain decreases in darting behavior. While the current work

provides a deeper understanding of generalized changes in PVN reactivity following psychedelic administration, a limitation of this study is that there were no cell-type specific approaches employed, and therefore specific mechanisms underlying changes in PVN reactivity remain unknown. Future work is needed to examine cell-type specificity in order to better characterize the alterations in reactivity seen in the PVN following psilocin administration.

All together, these data provide insight into the involvement of the PVN in psychedelic drug action. Given the relevance of HPA activation on psychiatric disease states, understanding how these drugs alter functioning within this stress-responding circuitry will provide key insights into better understanding how these drugs may elicit their therapeutic effects and provide potential predictive relevance in treatment responsiveness in human clinical populations.

## Methods

### Stereotaxic surgery
Male and female Sprague Dawley rats (200–400 g, ~8-weeks old at arrival, Envigo, Indianapolis, IN) were group-housed in a humidity- and temperature-controlled (22 °C) vivarium on a 12 hr light/dark cycle with *ad libitum* access to food and water. At ~9-weeks, animals were anesthetized using isoflurane with an initial bolus of 5%, followed by maintenance at 2–3%. Animals then underwent intracranial surgery

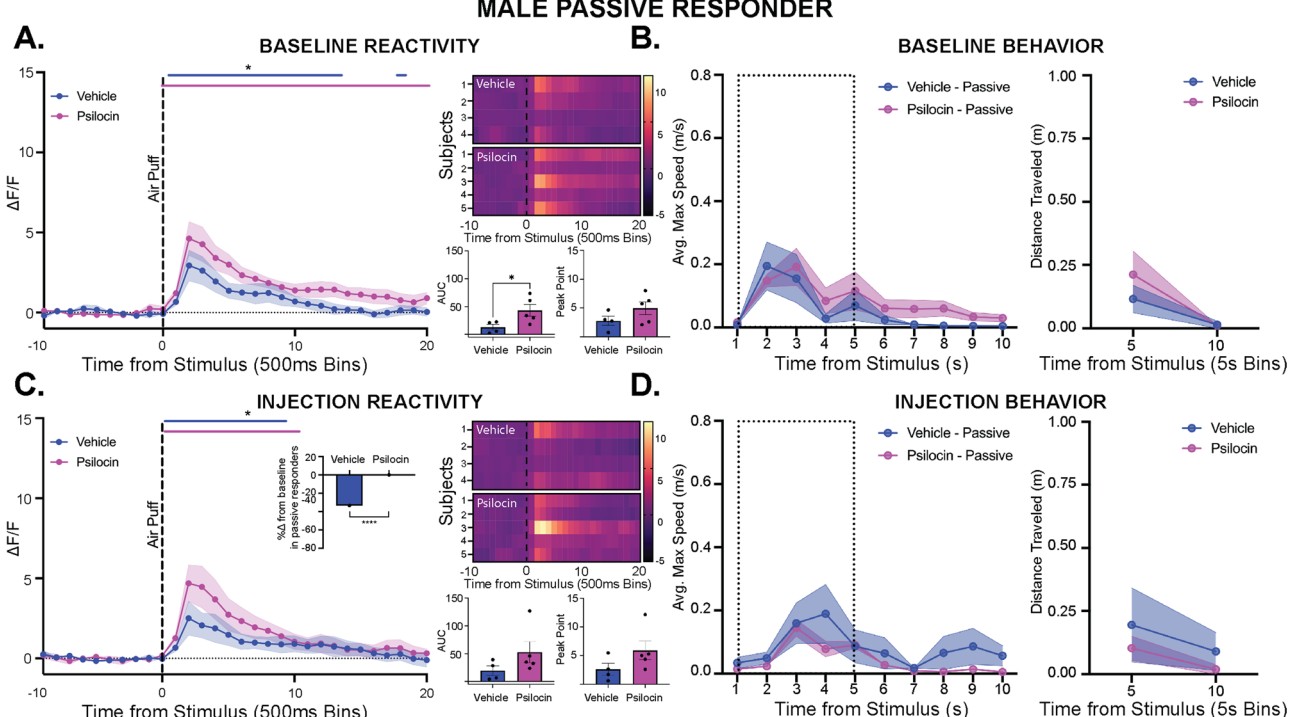

**Fig. 9 | Psilocin did not alter PVN reactivity in passive responding males.** Fiber photometry: Psilocin *n* = 5, Vehicle *n* = 4; Behavior: Psilocin *n* = 7, Vehicle *n* = 6; **A** Baseline PVN reactivity (left): ΔF/F trace plots of changes following exposure to a 500 ms air-puff. 2-way ANOVA was performed for statistical analysis. Data points represent group averages within 500 ms binned window +/- S.E.M. (shaded area); Heatmaps (top right) comparing individual responses to air-puff stimulus (dotted line) in vehicle and psilocin groups. Average AUC and PP +/- S.E.M (bottom right) compared by unpaired 2-tailed t-test between groups with a significant difference in AUC (p = 0.0468). Each data point represents an individual subject. **B** Average maximum speed (left) and distance traveled (right) following the air-puff stimulus. A 2-way ANOVA was performed for statistical analysis. Data points are group averages +/- S.E.M. (shaded area) (**C**) Day of injection PVN reactivity (left): ΔF/F trace plots of changes following exposure to a 500 ms air-puff. A 2-way ANOVA was performed for statistical analysis. Data points represent group averages within 500 ms binned window +/- S.E.M. (shaded area); Inset: Change in the proportion of passive responding animals compared to baseline (*p* < 0.0001) by 2-sided Fisher's exact test. The data point reflects the reduction in percentage from the baseline for the entire group. Heatmaps (top right) comparing individual responses to air-puff stimulus (dotted line) in vehicle and psilocin groups. Average AUC and PP +/- S.E.M. (bottom right) compared by unpaired 2-tailed t-test between groups. Each data point represents an individual subject. **D** Average maximum speed (left) and distance traveled (right) following the air-puff stimulus. A 2-way ANOVA was performed for statistical analysis. Data points are group averages +/- S.E.M. (shaded area). In each trace bin plot, a significant increase in ΔF/F was determined whenever the lower bound of the 99% CI was >0 with statistical significance shown as colored lines above each ΔF/F curve with colors corresponding to the respective binned traces *p < 0.05, **p < 0.01, ***p < 0.001, ****p < 0.0001. AUC = area under curve, PP = peak point, ΔF/F = change in fluorescence as a function of baseline fluorescence, CI = confidence interval.

using a (KOPF) stereotaxic frame. Bilateral infusions of pGP-AAV-syn-jGCaMP7f-WPRE (Addgene plasmid# 104488-AAV9) were performed at a rate of 100 nl/minute into the PVN using coordinates from bregma (AP: −1.8 mm, ML: +/− 0.5 mm, DV: −7.5 to −8.0 mm). Following injection, the needle was left in place for 5 minutes before retracting. Immediately following microinjections, three skull cap screws were put into place followed by implantation of a dual-tip fiber optic cannula (DFC_200/245-0.37_9mm_GS1.0_FLT), with fiber tips spaced 1 mm apart, placed at 0.3 mm above injection site (coordinates from bregma AP: −1.8 mm, ML: +/−0.0 mm, DV: −7.2 mm). Once fiber was in place, Metabond dental cement (Parkell, Brentwood, NY) was used to fix cannula in place and allowed to dry for 10 minutes before removing cannula holder and retracting stereotaxic arm. Animals were treated with 2.5 mg/kg of Flunixin on the day of surgery and for two days following. Following intracranial surgery at ~8 weeks of age, animals were single-housed in flat-lid cages with waterspout access, and food bowls to avoid damage to implants. All procedures adhered to the University of North Carolina Chapel Hill's Institutional Animal Care and Use Committee (IACUC).

### Drug Administration
On the day of injection, psilocin (Cayman Chemical) was dissolved in 2% glacial acetic acid and brought up to a 1 mg/ml dilution in 0.9% saline. Animals then received subcutaneous (s.c.) injections of either vehicle (2% glacial acetic acid in 0.9% saline) or psilocin and placed back into their home cage for 30 minutes prior to the beginning of habituation in open-field box. For the c-Fos cohort, animals received s.c. injections 2 h prior to perfusion and collection of brain tissue for immunohistochemical analysis.

### Fiber photometry recording
Animals underwent fiber photometry/behavioral recordings at ~12 weeks of age. All fiber photometry recordings were taken using the previously mentioned TDT RZ5 (Tucker-Davis Technologies, Alachua, FL) real-time processor and Doric external components and were also conducted in the same plexi-glass chamber[30]. To test how the PVN would respond to the air-puff stimulus, baseline recordings were taken capturing changes in fluorescence (ΔF/F) following air-puff administration. Animals were split into their respective groups (vehicle vs. psilocin) following baseline recordings, to ensure that baseline PVN reactivity was equivalent between the two groups at baseline. Cohorts were split into two subgroups and ran consecutively to ensure that all animals were running within the first half of the day. On each day of recording, animals were placed in the chamber for 10 minutes for habituation. On the day of injection, animals were given an injection of either vehicle or psilocin (2 mg/kg) and then placed back in their home

cage for 30 minutes prior to being connected to the fiber optic patch cord and the beginning of the habituation period. On the restraint stress days, animals first underwent a 20-minute restraint followed by a blood collection through tail bleed. Following these procedures, PVN reactivity was tested following the same air-puff parameters as mentioned above. For all fiber photometry experiments, the same male experimenter handled animals and recording. To avoid potential confounding effects of an added stressor, females were not checked for the estrous cycle.

## Air-puff stimulus

The air-puff stimulus[30] consisted of a custom made circuit controlling a solenoid that was connected to the air source within the room. Air pressure was regulated with a psi gauge and the solenoid was connected to the TDT system through a BNC cable. Activation of the solenoid through a button command on the TDT system opened the solenoid allowing for an 500 ms burst of air (85 psi) directed at the face of the animal. Tubing was connected to a metal pole and was inserted through a hole in the sound attenuated chamber containing the open field box environment that the animals were held in during recording sessions. This was manually lowered into the arena and aimed toward the animal's face before activating the circuit box and delivering an air-puff and simultaneous tagged event in the TDT system for peri-event signal analysis. Within each air-puff session, animals were exposed to a 500 ms 85 psi air-puff directed towards the face of the animal followed by a 5-minute inter-stimulus interval (ISI) and then a final repeated air-puff. For each animal, an average of the two air-puffs was collected and used as the reactivity readout.

## Locomotor behavior

Locomotor activity was assessed in a clear, plexiglass open field apparatus (50 cm×50 cm) which was located within a sound dampened activity cabinet. Behavioral sessions were recorded and analyzed with ANY-maze video tracking system (Stoelting Co, IL, USA). Behavioral measurements were collected within two time periods: a 10-minute habituation prior to the onset of the air puff and then a 5-minute ISI immediately following exposure to the air puff. Analyzes across time were calculated a 1 min resolution for the overall locomotor behaviors, and additionally at 1 s for the immediate post puff behavioral analysis to correspond with fiber photometry time points. Measures of locomotor activity included distance traveled (m), max speed (m/s), % time spent in the center zone, and time immobile (at 65% sensitivity, animal must remain immobile for at least 2 s). The center zone was defined within the tracking software ( ~ 7.6 cm×7.6 cm; 45% of total area), and the time spent variable was calculated as (time spent in center zone/ (total time spent in center + time spent in periphery)) × 100). A total of 8 subjects (3 females and 5 males) were excluded from behavioral analyses that included habituation time period (locomotion, time spent in center, immobility) due to technical difficulties with behavioral video recording.

## Restraint stress

Restraint stress procedures were performed using a Broome Rodent Restrainer (Braintree Scientific, Braintree MA). Animals were placed in a restraint stress tube with a foam insert to help protect against excessive bumping of the cannula on the plexiglass walls. Animals were restrained for 20 minutes. Following the 20-minute restraint, animals were kept in the tube for a 1–5 min for the collection of an ample volume of blood through the tail vein for cortisol analysis. Immediately following restraint stress and blood collection, animals were placed in the open field box for air-puff administration and PVN reactivity recording.

## Perfusion

Animals were first anesthetized using isoflurane prior to perfusion. Upon entry to the heart, the circulatory system was first flushed with phosphate-buffered saline (PBS) followed by at least 120 mL of 4% paraformaldehyde (w/v). Brains were then stored in 4% paraformaldehyde (w/v) for 24 h before being transferred to a 30% sucrose solution. Tissue was sectioned into 40 μm thick slices using a freezing microtome and then stored in a cryoprotectant solution (comprising 30% v/v ethylene glycol and 30% w/v sucrose in phosphate-buffered saline) at 4 °C until further use.

## Corticosterone assay

Tail blood was collected in 1.5 mL microcentrifuge tubes (Fisherbrand, Waltham, MA, CAT#05-408-129) and kept on ice. Blood samples were centrifuged, and serum was collected in 0.2 mL PCR tubes (Eppendorf, Enfield, CT, CAT#951010006) and stored at −20 °C. The serum was analyzed for corticosterone using a commercially available CORT ELISA kit (Arbor Assays, Ann Arbor, MI, CAT#K014-H5) that was performed per the manufacturer's protocol. Results from the corticosterone assay are expressed as nanograms/milliliter (ng/mL).

## Histology/Immunofluorescence staining

To assess changes in basal activity while under the acute effects of the drugs, animals in both groups were given subcutaneous injections of either vehicle or psilocin (2 mg/kg). Given that injections are inherently stressful, it is possible that both groups would have elevated levels of c-Fos expression within the PVN. While this could confound baseline levels of c-Fos, this perturbation is controlled between groups. Prior to injection, animals were perfused, and brain tissue was collected for analysis. 40 μm sections (AP coordinates −1.6 mm to −2.16 mm) were collected and prepared for immunohistochemistry (IHC) to stain for the protein c-Fos. The IHC protocol is as follows: wash in PBS, 30 min incubation in 50% methanol/PBS, 5 min in 3% hydrogen peroxide, blocking solution (0.3% Triton X-100; Thermo Fisher), 1 hr at room temperature (RT) in 1% bovine serum albumin, 24 hr incubation at 4 °C in rabbit anti-*cFos* (1:3000, Millipore Sigma; ABE457). Day 2: wash in 0.1% Tween-20 in tris-buffered saline (TNT), 30 min in TNB blocking buffer (Perkin-Elmer FP1012), 2 hr in goat anti-rabbit horseradish peroxidase (HRP; 1:200, Abcam ab6721), TNT washes, and finally a 10 min at RT incubation in tyramide conjugated fluorescein (1:50) in TSA amplification diluent (Akoya Biosciences, NEL741001KT). Following IHC, slices were mounted on slides using Vectashield ® HardSet™ Antifade Mounting Medium with DAPI (H1500, Vector Laboratories, Burlingame, CA). Images were acquired using a Keyence BZ-X800 fluorescence microscope. Images were taken at 20x magnification and the BZ-X800 Analyzer program was used to create stitched images for quantification. Quantification utilizing ImageJ[62] was conducted to manually count labeled, c-Fos+ cells. PVN region was outlined and counted by two separate experiments. The experimenter was blinded for counting. Counts were taken per hemisphere and averaged to obtain a value for each subject.

## Statistics and reproducibility

Collection of raw fiber photometry signal and subsequent data analysis to include binning of data into 500 ms time bins and bootstrapping confidence intervals were performed using Matlab (Version 2023a, The Mathworks, Inc.) and a custom written Matlab script (https://doi.org/10.5061/dryad.3ffbg79qr). For all binned trace data visualizations, we applied a bootstrapping confidence interval (CI) method (99% CI, 1000 iterations)[63]. This approach involved calculating a hypothetical 'mean' ΔF/F by repeatedly resampling the subject mean ΔF/F with replacement, aligning with the number of samples. The sample size (n) for all analyses matched the number of subjects per group. This bootstrapping procedure was performed 1000 times to generate a non-parametric distribution of population mean simulations for each time point within the window. A statistically significant increase (>0%) in Ca2+ transients following air-puff administration was identified by data points where the lower bound of the 99% CI exceeded 0[63–65] These

significant increases were indicated on graphs by color-coded lines above the trace bin plots. All comparisons of proportions were done using Fisher's exact tests. Comparisons of c-Fos, PP, AUC, and CORT concentrations were done using unpaired t-test to compare between groups at each time. 2-way RM ANOVAS were used to compare binned trace plots and behavioral data metrics between groups. All data were graphed and statistically analyzed with Prism 10.0.3 (GraphPad Software, MA, USA). For fiber photometry recordings, subjects were excluded from group data analysis if site verification revealed incorrect GCaMP injection location and/or fiber placement was outside of the PVN but was still included for behavioral analysis. Sample sizes for all experimental groups are listed within figure legends. Experimenters were blinded during manual cell counting and manual behavioral scoring. The experimenter was not blinded to condition during experimentation or during data analysis as these were all automated processes measuring changes in physiology and, therefore, did not involve any subjective interpretation during data collection. Sample sizes were determined based on previous work in the lab[30], however no a priori power calculations were conducted.

### Reporting summary

Further information on research design is available in the Nature Portfolio Reporting Summary linked to this article.

## Data availability

All data generated in this study, including a source data file containing raw data values for each figure, have been deposited in the Dryad database [https://doi.org/10.5061/dryad.3ffbg79qr]. Source data are provided in this paper.

## Code availability

Custom written MATLAB code and resulting MATLAB structures have been made available on the Dryad database [https://doi.org/10.5061/dryad.3ffbg79qr].

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

## Acknowledgements

This work was supported by National Institute of Health grants AA026858 (M.A.H.), GM135095 (D.P.E.), and by the Brain and Behavior Research Foundation NARSAD Young Investigator Award (M.A.H.)

## Author contributions

D.P.E. contributed to the initial conception, experimental design, apparatus design, surgical procedures, behavioral recording, histology, data analysis, custom code, data visualization, the initial draft of the manuscript, and manuscript editing. J.L.H. contributed to the behavioral recording, behavioral analysis, a section in methods, and manuscript editing. S.E.M. contributed to CORT assay, CORT analysis, and section in methods. S.N.M. contributed to CORT assay. S.G.Q. contributed to histological analyses, schematic illustrations, and manuscript editing. C.S.R. contributed to histology and draft editing. D.T. contributed to histology and manuscript editing. M.E.S. contributed to histology and manuscript editing. M.W.H. contributed to histology and manuscript editing. C.W.H. contributed to the behavioral recording and manuscript editing. M.A.H. contributed to the initial conception, experimental design, behavioral assay, and acquisition of funds.

## Competing interests

Authors have no competing interests to report.
