## [Peer Review File · Nature Communications]

REVIEWER COMMENTS

Reviewer #1 (Remarks to the Author):

The Authors are commended for their noteworthy effort to study an interesting molecule (psilocin) and its effect on cFos labelling and neural activity in the paraventricular nucleus of the thalamus (PVN), as well as its effect on PVN-associated behavior. The PVN is implicated in the regulation of emotional responses and may be an important brain target for psychiatric therapies. The study of the effect of psilocin on PVN and PVN-associated behavior is significant, if not only because little work has been done in this direction (drug and brain region combination). This research direction will likely become more significant in coming years as psilocybin and related molecules are likely to be decriminalized in several states and are likely to be FDA-approved for certain clinical indications.

Though this research direction is important, in the current manuscript the Authors appear to have stretched what is really just a couple experiments into an entire paper. The Authors do present convincing evidence that psilocin increases cFos and neural activity acutely in PVN. However, this result is buried by the many currently uninterpretable behavior data panels. Though statistically significant behavioral findings are presented the data appears highly variable and no power analysis are performed, making it not convincing. Figure 6 in particular is clearly underpowered.

As behavioral results constitute the majority of the data presented in the current manuscript, these experiments must be properly powered, which could be done in a revision with more batches of animals.

Reviewer #2 (Remarks to the Author):

This study aimed to examine the effect of one of psychedelic drug psilocin on PVN neuron reactivity during acute stress and restraint stress as well as associated behaviors. The results suggested that psilocin may cause reduced locomotion and increased responsiveness of PVN neurons with some variation between males and females. The effect of psychedelic drugs has gained a lot of traction as they produce some positive effects of psychiatric patients. In this sense, this study is significant. However, this study appears to be poorly designed and difficult to read. The results were not appealing and the conclusion was therefore not strong at all.

1) The conclusion of this study is not robust at all. The claimed changes in the PVN by the drug may have nothing to do with observed changes in responses of animals to stresses imposed and may only be remotely related to direct action of the drug. The observed effects may not be specific to PVN neurons at all, and the same or better responses may be obtained from some other brain regions.

2) The authors attempted to relate the findings to the HPA axis. However, no difference in Cort was identified. To make this relation stronger and more convincing, specific recording from PVN CRH neurons would be necessary.

3) For fiberphotometry studies, it is usually difficult to compare recordings between groups as the expression levels of the GCaMP virus may vary greatly between animals. Within this in mind, concerns were raised about data interpretation on results in most figures described in this study. Results in Fig. 6B seem to suggest that there is a baseline difference in GCaMP recordings, which demonstrates the concern.

4) The Fos result in Fig 1 is not convincing at all as the baseline Fos level in the PVN is so high. The authors are suggested to refer to literature on the baseline Fos level in rats, which normally shows a very low level of Fos expression. The high level of Fos at the baseline may suggest that the animal was stressed as opposed to baseline during the study.

5) For all study subjects, since the GCaMP virus is not Cre-dependent, the expression profile needs to be demonstrated to demonstrate specific expression in the PVN. The example picture shown in Fig.1, even with one section, suggests some level of expression outside PVN, which may confound data interpretation.

6) Results in Fig.4 are confusing: as depicted in Fig 1 C, the experiment was conducted at least more than 7 days after Psilocin administration. Since on day 7, results in other Figs already showed no effects from the drug, what is the purpose of this experiment?

7) The authors appear to argue that there was no difference between groups in Fig. 5I but there was difference between groups in Fig. 5K, which is not convincing at all.

8) In describing results, the authors should stay focused on the main findings and avoid those non-intended findings (i.e. there was an effect of time), which causes distractions in flow.

Reviewer #3 (Remarks to the Author):

In this study by Effinger et al., the authors aim to examine the effect of psychedelics on the behavioral and neural responses to stressors. Specifically, they investigate how psilocin affects PVN reactivity (as measured by in vivo fiber photometry recording of Ca⁺⁺) in response to aversive stimuli in male and female rats. They find that psilocin not only broadly affects PVN activity as measured by c-FOS, but also increases PVN reactivity in response to an air puff in male, but not female rats. This increase in PVN reactivity is transient, quickly returning to baseline when tested a few days later. Surprisingly, this male-specific increase in PVN reactivity is also predominantly found in males that respond to air puffs with an active threat response. The authors conclude that psilocin can alter PVN reactivity to an aversive threat stimulus in a sex-specific manner. Unfortunately I find that many of the findings in the paper are preliminary or lack novelty.

The main novel finding of the study is that psilocin administration prior to an aversive stimulus amplifies PVN activity in response to that aversive stimulus in males, but not females (Fig. 2D, Fig. 3D). This is presumably due to psilocin administered 30 min prior to the aversive stimulus enhancing PVN sensitivity (as shown by the increased c-FOS in Fig. 1A). This is interesting and is deserving of further study. For instance, why do females not show psilocin versus control differences in PVN responsiveness to the aversive stimulus even though they show an increase in PVN c-FOS in response to psilocin (Fig. 1B)? The authors argue that intra-animal variability may explain this difference (Fig. 3I, J) but this is not mechanistically explored. In fact, the standard deviation of the psilocin group's AUC and peak point responses on the day of injection in males (Fig. 2I, J) is, by eye, greater than (or at least not smaller than) that of females (Fig. 3I, J).

The other novel finding is that the large difference in PVN reactivity between control and psilocin-injected males can be explained by subdividing the mice into those that are "active responders" (dart or dash away from the aversive air puff) versus "passive responders" (immobility). The authors demonstrate that on the day of injection, PVN reactivity is much greater in psilocin-treated, active responding males versus control active responding males. Conversely, PVN reactivity does not significantly differ between psilocin-treated and control passive responding males. This is an exciting finding but based on the animal numbers in the active responding (4 control, 3 psilocin) and passive responding (2 control, 4 psilocin) groups, is preliminary. Furthermore, the authors propose an interesting potential mechanism for this difference in PVN reactivity based on threat response (Lines 423-435) but this is not mechanistically tested in the present manuscript.

It is also unclear to me why the authors tested the prolonged effects of psilocin on PVN reactivity 2 and 7 days after injection (Figs. 2F, 2H, 3F, 3H) given the short half-life of psilocin (50 min in plasma, DOI: 10.1080/03602532.2016.1278228). Because of this it is unsurprising that there is no difference PVN reactivity to restraint stress in control versus psilocin-treated rats (Figs. 4B, D) because the restraint stress happens sometime after 7 days after injection (it is unclear from the schematic in Fig. 1C and from the methods).

Other claims made by the authors lack novelty, for example: it is unsurprising that psilocin decreases general locomotor activity (Fig. 1C, DOI: 10.1097/FBP.000000000000198 and others), and it is unsurprising that corticosterone responses to restraint stress are greater in females than in males (Fig. 4E, DOI: 10.3109/10253890.2013.777832 and many others).

REVIEWER COMMENTS

Reviewer #1 (Remarks to the Author):

The Authors are commended for their noteworthy effort to study an interesting molecule (psilocin) and its effect on cFos labelling and neural activity in the paraventricular nucleus of the hypothalamus (PVN), as well as its effect on PVN-associated behavior. The PVN is implicated in the regulation of emotional responses and may be an important brain target for psychiatric therapies. The study of the effect of psilocin on PVN and PVN-associated behavior is significant, if not only because little work has been done in this direction (drug and brain region combination). This research direction will likely become more significant in coming years as psilocybin and related molecules are likely to be decriminalized in several states and are likely to be FDA-approved for certain clinical indications.

Though this research direction is important, in the current manuscript the Authors appear to have stretched what is really just a couple experiments into an entire paper. The Authors do present convincing evidence that psilocin increases cFos and neural activity acutely in PVN. However, this result is buried by the many currently uninterpretable behavior data panels. Though statistically significant behavioral findings are presented the data appears highly variable and no power analysis are performed, making it not convincing. Figure 6 in particular is clearly underpowered.

As behavioral results constitute the majority of the data presented in the current manuscript, these experiments must be properly powered, which could be done in a revision with more batches of animals.

1. “However, this result is buried by the many currently uninterpretable behavior data panels.”

We appreciate the reviewer’s comment. This comment illuminated an unintentional focus on broad behavioral results, rather than the intention of providing real time PVN reactivity corresponding to behavioral outputs. To this end, we have implemented substantial changes to the figure layout and results. Figures 2 and 3 now contain only the threat responding behavior corresponding to the window of time for the stimulus presentation during fiber recordings. All habituation and post-stimulus locomotor and center time data have been moved to supplement. We believe this comment was invaluable in that it provided an opportunity to make changes to the layout of the manuscript in a way that more accurately conveys the most compelling aspects of this work and increases the clarity of communication.

2. “Though statistically significant behavioral findings are presented the data appears highly variable and no power analysis are performed”

Sample sizes were determined based on previously published work. Language has been added to results to clarify this point. While we initially performed surgery on enough animals to yield sample sizes of 12 subjects, the use of a dual tip fiber cannula in conjunction with surgical complications involved in the approach for recording from a region as deep and small as the PVN brought our total animal numbers down. Considering these concerns, we have added animals to our groups to increase sample sizes (new sample sizes: male psilocin: n = 9; male vehicle: n = 9; female psilocin: n = 8; female vehicle: n = 8). Attached are observed power calculations for the area under the curve (AUC) fiber photometry measures for the males on the day of injection, where we observed significant effects that fulfill the 80% threshold criterion.

Comparing Injection Fiber Photometry between groups

AUC

Study Parameters	
Mean, group 1	70.65
Mean, group 2	25.67
Subjects, group 1	9
Subjects, group 2	9
Alpha	0.05

$$\text{Power} = \Phi \left\{ -Z_{1-\alpha/2} + \frac{\Delta}{\sqrt{\sigma_1^2/n_1 + \sigma_2^2/n_2}} \right\}$$
$$\text{Power} = \Phi \left\{ -(1.96) + \frac{44.98}{\sqrt{41.5^2/9 + 24.54^2/9}} \right\}$$
$$\text{Power} = \Phi \left\{ 0.839 \right\} = 0.799 = 79.9\% \text{ power}$$

Post-hoc Power
79.9% power

3. "Figure 6 in particular is clearly underpowered."

Figure 6 included the classification of experimental subjects by baseline threat responding behavior, which is unfortunately a parameter that is outside of our direct control. While we agree that having more subjects for baseline behavior subgroups in our initial submission would have been ideal, the splitting of the males into subgroups occurred after the initial experimental cohort was completed. We have subsequently added more subjects to our groups to address this concern (New sample sizes: active male psilocin: n = 4; active male vehicle: n = 5; passive male psilocin: n = 5; passive male vehicle: n = 4). Attached is an observed power calculation for AUC of the fiber photometry traces on the day of injection comparing active psilocin vs active vehicle, where we found the effect. As is evident in the included graphic, with the updated sample sizes, we get observed power above the 80% preferred criterion.

Active Responding Males Fiber Photometry

AUC

Study Parameters	
Mean, group 1	30.52
Mean, group 2	92.98
Subjects, group 1	5
Subjects, group 2	4
Alpha	0.05

$$Power = \Phi \left\{ -Z_{1-\alpha/2} + \frac{\Delta}{\sqrt{\sigma_1^2/n_1 + \sigma_2^2/n_2}} \right\}$$

$$Power = \Phi \left\{ -(1.96) + \frac{62.46}{\sqrt{30.36^2/5 + 32.11^2/4}} \right\}$$

$$Power = \Phi \{ 1.011 \} = 0.844 = 84.4\% \text{ power}$$

Post-hoc Power	
84.4%	power

4. "these experiments must be properly powered, which could be done in a revision with more batches of animals."

We have added more subjects to both males and females to increase the sample sizes for all experimental groups and address these concerns of the reviewer. Additionally, we have performed the attached observed power calculations to further substantiate the sufficiency of our experimental groups, which we hope satisfies the concerns regarding sample size.

Reviewer #2 (Remarks to the Author):

This study aimed to examine the effect of one of psychedelic drug psilocin on PVN neuron reactivity during acute stress and restraint stress as well as associated behaviors. The results suggested that psilocin may cause reduced locomotion and increased responsiveness of PVN neurons with some variation between males and females. The effect of psychedelic drugs has gained a lot of traction as they produce some positive effects of psychiatric patients. In this sense, this study is significant. However, this study appears to be poorly designed and difficult to read. The results were not appealing and the conclusion was therefore not strong at all.

1) The conclusion of this study is not robust at all. The claimed changes in the PVN by the drug may have nothing to do with observed changes in responses of animals to stresses imposed and may only be remotely related to direct action of the drug. The observed effects may not be specific to PVN neurons at all, and the same or better responses may be obtained from some other brain regions.

The goal of this work was to examine the effects of psychedelic administration on reactivity within the PVN and threat responding behavior. While human imaging has provided insight into changes in functional connectivity, this work sought to incorporate the strengths of a preclinical approach to investigate an integral nucleus within the hypothalamus that human imaging lacks the resolution to specifically investigate. While we did explore functional changes using aversive stimuli and collected measurements of stress-related metrics like elevated corticosterone, the PVN is a highly heterogenous nucleus within the hypothalamus and plays a key role in a variety of different functions outside of stress-responding, including autonomic function, appetitive behavior, and social behavior. Given that the PVN is involved in affective processing and social behaviors broadly, and that PVN dysregulation is

observed in many psychiatric disorders, there was strong rationale in exploring the effects of psychedelics in this region independent of stress responding. Given the considerable gap in knowledge concerning region-specific effects of psychedelic drug actions, our approach was to assess *the primary* changes in PVN activity using immunohistochemistry as well as genetically encoded calcium sensors to measure changes in baseline activity as well as stimulus-response reactivity. The insight gained from this study will inform more directed and specific interrogation into the cell type-specific effects of psychedelics on PVN activity at rest and in different stress conditions. Regarding the concern suggesting that the observed effects may not be related to direct actions of the drug, this study included implementation of a vehicle control group wherein all timepoints and conditions were tightly controlled for such that any changes observed are most likely due to effects of the experimental drug. This approach is consistent with previous work (Effinger et al., 2023), and the current study is part of an accumulation of knowledge concerning brain region-specific effects of psychedelic drug action. Language has been added to the manuscript to acknowledge the scope and limitations of the current study and ensure that all interpretation and conclusions are supported by experimental data (**pg. 9 line 27-pg 10 line 3; pg.14 line 1-4**).

2) The authors attempted to relate the findings to the HPA axis. However, no difference in Cort was identified. To make this relation stronger and more convincing, specific recording from PVN CRH neurons would be necessary.

The reviewers concern regarding CORT levels and HPA axis activity is valid, however, we may not have clearly communicated the rationale for the acute stress and CORT measurements within this study. While the PVN is indeed an integral, hub region within the HPA axis, the purpose of this study was not to interrogate the effects of psychedelic drugs on HPA reactivity as determined by CORT levels. Previous work from a group at University of Wisconsin already demonstrated that psilocybin produces acute increases in CORT in mice (Jones et al., 2023). As previously stated, the PVN is a highly heterogeneous region and our interest in PVN involvement in psychedelic drug action was not solely due to its involvement in CRF release and downstream activation of the endocrine stress response, but as a central brain region involved in a number of physiological processes. In this study, serum CORT levels were analyzed following restraint stress to provide evidence that the 20-minute restraint stress was sufficient to drive a peripheral stress response. Given that animals had previous exposure to psilocin, we were also able to investigate the effects of that prior exposure on CORT release following acute stress, however that was not the point of that assay in this context. Given that we are using this restraint stress assay as an acute stressor prior to testing PVN reactivity with the air-puff, we felt it was necessary to show that we were indeed driving elevated levels of CORT. Language has been added to the results (**pg.7 line 3-6**) and discussion section (**pg. 9 line 27-pg 10 line 3; pg. 10 line 18-21; pg.14 line 1-4**) of the revised manuscript to clarify this point. We also acknowledged that non-specific cell targeting is a limitation of this study within the discussion and that changes in PVN reactivity following psychedelic administration could be due to cell-specific changes and is the focus for future studies. While the actions of psychedelics specifically on PVN CRH neurons is an excellent area of inquiry, the cell-type specific approach required for this undertaking would more than double experimental group numbers and is outside the scope of the current study. Future work will absolutely build on this, and the current work will provide key foundational information to aid in the development of more informed, hypothesis-driven approaches to explore the PVN's involvement in psychedelic drug action.

3) For fiber photometry studies, it is usually difficult to compare recordings between groups as the expression levels of the GCaMP virus may vary greatly between animals. With this in mind, concerns were raised about data interpretation on results in most figures described in this study. Results in Fig. 6B seem to suggest that there is a baseline difference in GCaMP recordings, which demonstrates the concern.

The reviewer brings up a good point. One important consideration of studies utilizing genetically encoded sensors is the potential for differential expression that could potentially drive differences in fluorescence between subjects. The current study controls for this, as groups were assigned *following* baseline recording days. Metrics such as area under the curve and peak point were used to ensure that we had equivalent baseline responding between the vehicle control and psilocin groups prior to drug administration. The reviewer refers to a baseline discrepancy seen between groups in figure 6, however this figure includes behavioral subgroups that were assigned *posthoc*. Given that these groups were determined after the experiment had been performed, we were not able to control for baseline reactivity in the active and passive responders. Additionally, these differences may provide context to better understand these potential behavioral phenotypes, in that active responders may have greater PVN reactivity at baseline than passive responders, which would be very interesting. It should be noted that, when combined into the overall male group, there are no differences in baseline responding between vehicle and psilocin groups. Language has been added to the text in the results (**pg.4 lines 16-18**) and methods (**pg.15 line 13-15**) sections clarifying our strategy in assigning groups and controlling for equivalent levels of reactivity at baseline between groups.

4) The Fos result in Fig 1 is not convincing at all as the baseline Fos level in the PVN is so high. The authors are suggested to refer to literature on the baseline Fos level in rats, which normally shows a very low level of Fos expression. The high level of Fos at the baseline may suggest that the animal was stressed as opposed to baseline during the study.

While the PVN is a relatively small region, you can see stark differences in the density of the cell population as you move from anterior to posterior. For instance, Yamaguchi et al. report 139 ± 29 c-Fos+ cells in their control [1]. Additionally, Weiser et al. show c-Fos+ cells to be around 150 in the vehicle control group in a PVN microinjection study [2]. While both of these studies show lower levels of c-Fos+ cells under non-stressed conditions, our current study necessitates a subcutaneous injection to determine drug effects that would align with the administration strategy used in the in-vivo recording experiments. Injections are inherently stressful, so as a result, we would expect this to have some effect on c-Fos expression, however this is controlled for as both groups received the same volume of injection, through the same route of delivery, from the same experimenter. The reviewer brings up a good point, as it seems some key aspects of our approach were left out of the methods section that would address these concerns. Language has been added to results (**pg.4 lines 3-5**) and methods (**pg.18 line 1-5**) to address these concerns of the reviewer.

5) For all study subjects, since the GCaMP virus is not Cre-dependent, the expression profile needs to be demonstrated to demonstrate specific expression in the PVN. The example picture shown in Fig. 1, even with one section, suggests some level of expression outside PVN, which may confound data interpretation.

The reviewer brings up a fair point concerning possible spread of GCaMP expression outside of the confines of the PVN. However, the figure that they are referring to is actually a representative image of c-Fos expression, not GCaMP (two different cohorts). However, to address the concern of the reviewer, within fiber photometry studies, there is often some spread of virus outside of the region of interest, especially in regions as small as the PVN. However, site verification for fiber photometry experiments places much more emphasis on placement of the fiber. For our studies, we utilized a dual-tip fiber optic cannula with a numerical aperture of 0.37. This refers to the angle of the light at the tip of the fiber (see image below). As you can see, 0.37 NA offers a relatively confined area of illumination, and therefore restricts the area from which GCaMP fluorescence can be collected. During site verification, subjects were excluded if GCaMP was not seen in the PVN or if the fiber placement was not directly within the PVN. So, while there may have been some spread outside of the PVN for GCaMP expression, we

confirmed that all included subjects had correct placement of the fiber and GCaMP expression to ensure that recordings were being collected from the correct area.

6) Results in Fig.4 are confusing: as depicted in Fig 1 C, the experiment was conducted at least more than 7 days after Psilocin administration. Since on day 7, results in other Figs already showed no effects from the drug, what is the purpose of this experiment?

The reviewer's comment illustrates a missed opportunity for us to adequately explain the rationale for our approach in this experiment. Psychedelic compounds, such as psilocin, have received recent renewed interest as potential treatments for psychiatric disorders. One noteworthy feature of these compounds is they appear to elicit rapid acting and persistent therapeutic effects, with effects being seen to last for days and months following a single administration. Importantly, the underlying neurobiological mechanism mediating these persistent effects remains unknown. Therefore, investigation into the more prolonged effects of these drugs on brain region reactivity in preclinical models is incredibly relevant. In fact, previously published work in the lab demonstrated a prolonged decrease in reactivity within the central amygdala, seen as early as 2 days, but lasting as far out as 28 days post-injection of the drug, effects that were not seen in vehicle control or female subjects (Effinger et al., 2023). Language explaining the rationale for recordings at more prolonged time points has been added to the revised manuscript to clarify the point of these studies (**pg.5 line 16; pg.7 line 1-2**).

7) The authors appear to argue that there was no difference between groups in Fig. 5I but there was difference between groups in Fig. 5K, which is not convincing at all.

In response to concerns from reviewer 1, we have added subjects to this experiment in order to increase sample sizes and power. This was especially important for the posthoc separation of males into behavioral subgroups based on their threat responding behavior exhibited at baseline. Following addition of new subjects, the discrepancy between Fig. 5I (now Fig.6D) and 5K (now Fig.6H) has been resolved. The differences seen between groups in Fig.5K (now figure 6H) no longer remains.

8) In describing results, the authors should stay focused on the main findings and avoid those non-intended findings (i.e. there was an effect of time), which causes distractions in flow.

This was a great suggestion from the reviewer. In describing the data, we realized that stating the redundant effect of time was distracting readers from the primary findings of this paper. In response to this comment and the comments of reviewer 1, we have substantially revised the structure of the figures and results, which included putting the locomotion and time spent in center plots into supplement. We feel that as a result of this suggestion, the paper has been drastically improved and allows readers to better appreciate the novelty of this approach and key findings. By placing threat responding behavior alongside the PVN fiber photometry data corresponding to the exact same window of time, we can allow more focus on the central findings of this paper with less distraction.

Reviewer #3 (Remarks to the Author):

In this study by Effinger et al., the authors aim to examine the effect of psychedelics on the behavioral and neural responses to stressors. Specifically, they investigate how psilocin affects PVN reactivity (as measured by in vivo fiber photometry recording of Ca^{++}) in response to aversive stimuli in male and female rats. They find that psilocin not only broadly affects PVN activity as measured by c-FOS, but also increases PVN reactivity in response to an air puff in male, but not female rats. This increase in PVN reactivity is transient, quickly returning to baseline when tested a few days later. Surprisingly, this male-specific increase in PVN reactivity is also predominantly found in males that respond to air puffs with an active threat response. The authors conclude that psilocin can alter PVN reactivity to an aversive threat stimulus in a sex-specific manner. Unfortunately I find that many of the findings in the paper are preliminary or lack novelty.

The main novel finding of the study is that psilocin administration prior to an aversive stimulus amplifies PVN activity in response to that aversive stimulus in males, but not females (Fig. 2D, Fig. 3D). This is presumably due to psilocin administered 30 min prior to the aversive stimulus enhancing PVN sensitivity (as shown by the increased c-FOS in Fig. 1A). This is interesting and is deserving of further study. For instance, why do females not show psilocin versus control differences in PVN responsiveness to the aversive stimulus even though they show an increase in PVN c-FOS in response to psilocin (Fig. 1B)? The authors argue that intra-animal variability may explain this difference (Fig. 3I, J) but this is not mechanistically explored. In fact, the standard deviation of the psilocin group's AUC and peak point responses on the day of injection in males (Fig. 2I, J) is, by eye, greater than (or at least not smaller than) that of females (Fig. 3I, J).

The other novel finding is that the large difference in PVN reactivity between control and psilocin-injected males can be explained by subdividing the mice into those that are "active responders" (dart or dash away from the aversive air puff) versus "passive responders" (immobility). The authors demonstrate that on the day of injection, PVN reactivity is much greater in psilocin-treated, active responding males versus control active responding males. Conversely, PVN reactivity does not significantly differ between psilocin-treated and control passive responding males. This is an exciting finding but based on the animal numbers in the active responding (4 control, 3 psilocin) and passive responding (2 control, 4 psilocin) groups, is preliminary. Furthermore, the authors propose an interesting potential mechanism for this difference in PVN reactivity based on threat response (Lines 423-435) but this is not mechanistically tested in the present manuscript.

It is also unclear to me why the authors tested the prolonged effects of psilocin on PVN reactivity 2 and 7 days after injection (Figs. 2F, 2H, 3F, 3H) given the short half-life of psilocin (50 min in plasma, DOI: 10.1080/03602532.2016.1278228). Because of this it is unsurprising that there is no difference PVN reactivity to restraint stress in control versus psilocin-treated rats (Figs. 4B, D) because the restraint stress happens sometime after 7 days after injection (it is unclear from the schematic in Fig. 1C and from the methods).

Other claims made by the authors lack novelty, for example: it is unsurprising that psilocin decreases general locomotor activity (Fig. 1C, DOI: 10.1097/FBP.000000000000198 and others), and it is unsurprising that corticosterone responses to restraint stress are greater in females than in males (Fig. 4E, DOI: 10.3109/10253890.2013.777832 and many others).

1. why do females not show psilocin versus control differences in PVN responsiveness to the aversive stimulus even though they show an increase in PVN c-FOS in response to psilocin (Fig. 1B)? The

authors argue that intra-animal variability may explain this difference (Fig. 3I, J) but this is not mechanistically explored. In fact, the standard deviation of the psilocin group's AUC and peak point responses on the day of injection in males (Fig. 2I, J) is, by eye, greater than (or at least not smaller than) that of females (Fig. 3I, J).

Given that c-Fos expression is a marker of neuronal activity, we utilized labeling of c-Fos+ cells following administration of the drug to capture differences in basal activity within the PVN. Here we show, that in both males and females, basal activity as measured by c-Fos expression was increased following administration of psilocin. The fact that we don't see the same effect in the fiber photometry data in females is suggestive of potential sex-differences in stimulus-induced activation. As opposed to basal activation, stimulus-induced changes can be impacted by a variety of different mechanisms and could point towards differential cell-type expression or innervating circuitry between sexes that could govern differential stimulus-induced activation.

To the reviewers point regarding intra-animal variability in females, this was an initial concern because there were increases in AUC and PP only seen in 1 or 2 subjects that seemed to drive the effect. After adding subjects to this study, this increase was no longer apparent. The text has been adjusted to remove statements regarding differences in females and variability. Additionally, we have removed within-subject comparisons as it appeared to provide more confusion than clarity for the reader. Given that baseline responding was controlled for, we feel comparison between groups across different time points is the most effective way of exploring drug effects in this study.

2. "This is an exciting finding but based on the animal numbers in the active responding (4 control, 3 psilocin) and passive responding (2 control, 4 psilocin) groups, is preliminary. Furthermore, the authors propose an interesting potential mechanism for this difference in PVN reactivity based on threat response (Lines 423-435) but this is not mechanistically tested in the present manuscript.

In response to the concern of the reviewers (see Reviewer 1: comment 3 and 4) regarding low sample sizes in the active vs. passive responders, more subjects have been added to the current study to increase all experimental groups. The difference in PVN reactivity by baseline threat responding behavior was an unanticipated and novel finding that will inform all future studies, and therefore the ability to control for sample sizes demonstrating these divergent behaviors was outside of our control. Following the addition of the new subjects, we have a strong observed power in the effects seen in the active responders (see Rev 1, comment 3) and feel very confident that these findings are in fact real. More work will be done to build on these findings, explore these behavioral phenotypes, and a potential synergistic effect with psychedelic drug action. Language has been added to results to soften our interpretation of these findings and acknowledge the limitation of the current study (**pg.13 line 3-5**).

3. "It is also unclear to me why the authors tested the prolonged effects of psilocin on PVN reactivity 2 and 7 days after injection (Figs. 2F, 2H, 3F, 3H) given the short half-life of psilocin (50 min in plasma, DOI: 10.1080/03602532.2016.1278228). Because of this it is unsurprising that there is no difference PVN reactivity to restraint stress in control versus psilocin-treated rats (Figs. 4B, D) because the restraint stress happens sometime after 7 days after injection (it is unclear from the schematic in Fig. 1C and from the methods).

The reviewer's concern is also shared with reviewer 2 (See Reviewer 2: comments 2 and 6), highlighting the need to adequately explain our rationale for exploring prolonged time points. Psychedelic drugs have received renewed interest in part, due to rapid-acting and persistent effects following a single dose. In previously published work, we showed that a single administration of psilocin produced reductions in central amygdala reactivity at 2-, 6-, and 28-days post administration (Effinger et al., 2023). Following the discovery of these persistent effects in the CeA,

we designed this experiment to investigate potential prolonged effects within the PVN. Given that participants in clinical trials report persistent reductions in symptoms related to anxiety and depression, we utilized a restraint stress on day 8 followed by testing PVN reactivity to determine how the PVN may function following an acute stressor, and in doing so determine if there were any prolonged effects in PVN reactivity that may have emerged through altered stress-related reactivity. Our thought was that this could provide converging evidence towards gaining a better understanding of the persistent therapeutic effects seen in the clinic. Language has been added to better explain the rationale for prolonged time points and for investigating acute stress exposure in assessing the effects of psilocin on PVN reactivity and threat responding behavior (**pg.5 line 16; pg.7 line 1-2; pg.7 line 3-6**).

4. Other claims made by the authors lack novelty, for example: it is unsurprising that psilocin decreases general locomotor activity (Fig. 1C, DOI: 10.1097/FBP.0000000000000198 and others), and it is unsurprising that corticosterone responses to restraint stress are greater in females than in males (Fig. 4E, DOI: 10.3109/10253890.2013.777832 and many others).

The reviewer points out an unintentional focus on psychedelic induced decreases in locomotor activity as a central finding. We agree that the decreases in locomotion by psilocin are not novel, however we do feel that inclusion of generalized behavior such as locomotion is necessary in any behavioral neuropharmacology experiment, so have reformatted the figures and corresponding results to move the more general behavioral findings to supplement. Additionally, our finding that females showed greater corticosterone response was not meant to be portrayed as novel, but rather is in alignment with the literature and therefore compliments the intended purpose of the CORT assay. Given that we were utilizing a restraint stress paradigm as an acute stressor prior to PVN recording, we wanted to assess CORT levels as a way of validating that we were able to induce a stress response. Language has been added to signify that these data replicate previous findings and are therefore not novel (**pg.7 line 10-12**).

1. Yamaguchi, N., et al., *The combination of cholecystokinin and stress amplifies an inhibition of appetite, gastric emptying, and an increase in c-Fos expression in neurons of the hypothalamus and the medulla oblongata*. *Neurochemical Research*, 2020. **45**(9): p. 2173-2183.
2. Weiser, M.J., C. Osterlund, and R.L. Spencer, *Inhibitory effects of corticosterone in the hypothalamic paraventricular nucleus (PVN) on stress-induced adrenocorticotrophic hormone secretion and gene expression in the PVN and anterior pituitary*. *Journal of neuroendocrinology*, 2011. **23**(12): p. 1231-1240.

REVIEWER COMMENTS

Reviewer #1 (Remarks to the Author):

The authors are to be commended for their improved manuscript investigating the effect / role of psilocin on affective behavior potentially through the paraventricular nucleus (PVN). This is important as psychedelics are likely to be decriminalized and used as treatments in the future, and it is unclear how their therapeutic effects work through the neural circuitry. The usual critique that the authors have not manipulated the PVN to test causality is somewhat more relevant here as the PVN is highly connected, and may merely relay behavioral information from one brain center to another.

Regardless, upon being confronted with the many behavioral charts and graphs comparing males and females behaviorally, I decided to look at them myself to see what results I came away with. Upon doing so in Figure 1 I was impressed that psilocin appears to increase cFos staining in the PVN for both males and females. In Figure 2, it seems as though psilocin increases reactivity in PVN, although that cohort of animals is trending toward having higher reactivity in PVN anyhow. In Figure 3C-D, having already seen Figure 2C-D, I am more encouraged that psilocin increases reactivity in PVN -- psilocin also appears to reduce locomotion. *If males and females were combined, the C-D's would be more clear and convincing. In Figure 4 corticosterone is clearly higher in females very convincingly. In Figure 5 it is quite convincing that active responders are decreased after psilocin for males and females. In Figure 6 there are still quite few animals although the authors have performed a power analysis -- I still wonder how much of the differences in basal responses with these cohorts are influencing the result.

Taken together "Psilocin increases activity in PVN which is associated with reduced active threat response". This is very cool! and maybe should be the title which is somewhat uninformative currently.

Thus, my feeling is that there are very strong results in this manuscript but that it is still difficult to uncover them with the many male and female comparisons and the repetitive nature of the writing (particularly the "Acute and prolonged effects of psilocin on PVN reactivity and behavior in males" then "...females" sections). I think this paper would be much stronger and perhaps ready for publication with just 4 figures if male and female data were combined for all experiments (except for corticosterone perhaps) -- Current graphs can just be cloned into the supplement, and combined graph presented as main figures.

Reviewer #1 (Remarks on code availability):

n

Reviewer #2 (Remarks to the Author):

I would like to thank the reviewers for the efforts made to address the concerns raised in the previous review. However, for this reviewer, the quality of data and conclusion drawn from the current study won't still merit the publication in Nat Comm.

On the point of data quality, 1) for the c-Fos data, the authors argued with citation with some previous studies; however, the current studies showed extraordinary number of baseline c-Fos expression (I would guess each section would have more than 200 neurons positive). In addition, if any previous studies showing a low level of c-Fos at baseline, the current study should have that baseline expression of expression. Please see PMID 26412230 for the level of c-Fos expression at baseline at per section level. Bad experiments in previous publications couldn't justify this study here. Also the core data presented here solely relying on fiberphotometry with a mild difference to support the conclusion are not robust.

On the significance part, all data presented here are based on correlation between activity and locomotion. There are no data that could show a causal role for PVN neurons in contributing the drug effects on behavior.

In addition, although it is true the psychedelics drugs produce some promising effects on psychiatric behaviors, none of the current experiments showed any effects on the beneficial effects by the used psychedelics drug. The significance of the changes in locomotion is not clear.

The data on the difference between males and females and between male aggressor and non-aggressor appear to be more interesting. Unfortunately no mechanistic studies were provided.

Reviewer #2 (Remarks on code availability):

I am not familiar with coding and couldn't provide an assessment on this.

Reviewer #3 (Remarks to the Author):

In this revised manuscript, the authors have increased their ns for many of their experiments and moved many of the behavioral findings to the supplemental. The main finding of the manuscript are that psilocin acutely increases PVN activity in response to an air puff in "active threat responding" male rats, but not passive threat responding male rats or female rats. Many of the findings are negative results. This is not necessarily a bad thing but is counter to mentions throughout the manuscript of the prolonged effects of psilocin on behavior and neuronal activity (i.e. days or weeks after administration). If psilocin only acutely shows a difference in PVN reactivity, it raises the question of the relevance of the PVN for psilocin's persistent effects on behavior.

The authors' responses to my previous comments are, for the most part, acceptable. The argument that the male-female increase in c-FOS staining in the PVN after psilocin injection seen in Fig. 1 reflects "basal" PVN activity is somewhat weak. The authors admit that injections themselves are stressors (lines 54, 55), so their c-FOS staining results are not measuring basal c-FOS levels. This changes the interpretation of their results: the stressful stimulus of psilocin injection increases PVN activity above that of the stressful stimulus of vehicle injection in both males and females. The change in PVN activity due to the air puff is due to a second stressful stimulus that seems to differentially affect males and females.

I would also recommend measuring changes in PVN reactivity using a two-way ANOVA versus just measuring the AUC or peak point. By eye, the female psilocin PVN response to an air puff in Fig. 3C looks to be overall greater than the female vehicle PVN response. It might not be statistically significant but I'd recommend checking.

REVIEWER COMMENTS

Reviewer 1

1. The authors are to be commended for their improved manuscript investigating the effect / role of psilocin on affective behavior potentially through the paraventricular nucleus (PVN). This is important as psychedelics are likely to be decriminalized and used as treatments in the future, and it is unclear how their therapeutic effects work through the neural circuitry. The usual critique that the authors have not manipulated the PVN to test causality is somewhat more relevant here as the PVN is highly connected, and may merely relay behavioral information from one brain center to another. The usual critique that the authors have not manipulated the PVN to test causality is somewhat more relevant here as the PVN is highly connected, and may merely relay behavioral information from one brain center to another.

We thank the reviewer for the positive perspective on our work. The reviewer brings up a valid point regarding the importance of investigating causal effects of PVN activity and engagement of relevant PVN circuits in key behavioral readouts. The current study provides valuable and novel insights into PVN function and a potential involvement of the PVN in the acute effects of psychedelic drugs that can be used to develop future studies investigating causal roles and circuit contributions. Text was previously added to the revised discussion to address this limitation and **additional text has been added to the discussion (pg. 10 line 8-9).**

2. Regardless, upon being confronted with the many behavioral charts and graphs comparing males and females behaviorally, I decided to look at them myself to see what results I came away with. Upon doing so in Figure 1 I was impressed that psilocin appears to increase cFos staining in the PVN for both males and females. In Figure 2, it seems as though psilocin increases reactivity in PVN, although that cohort of animals is trending toward having higher reactivity in PVN anyhow. In Figure 3C-D, having already seen Figure 2C-D, I am more encouraged that psilocin increases reactivity in PVN -- psilocin also appears to reduce locomotion. *If males and females were combined, the C-D's would be more clear and convincing. In Figure 4 corticosterone is clearly higher in females very convincingly. In Figure 5 it is quite convincing that active responders are decreased after psilocin for males and females. In Figure 6 there are still quite few animals although the authors have performed a power analysis -- I still wonder how much of the differences in basal responses with these cohorts are influencing the result.

The reviewer brings up important points regarding basal responses and the potential value of combining sexes to increase overall group numbers and the power of our findings. To investigate these concerns, we first performed a comparison between males and females at baseline and, as noted by the reviewer, there is increased basal activation in the males compared to females. The significant difference in baseline reactivity precludes combining males and females for the rest of the experimental analysis, therefore we don't feel that combining males and females would be beneficial to the communication of the data. **This information has been added to Figure 1 and in the results (pg. 5 line 1-4) and discussion (pg.10 line 14-15).** For what it's worth, we did perform the combined analysis, and the effect remained, but given that the

females do not show a significant increase in reactivity following psilocin injection, we feel that combining them would be misleading and an inaccurate representation of the findings. This was an invaluable comment as we feel the inclusion of the sex-differences at baseline add another novel finding to this paper and strengthen its impact. The reviewer also brings up concern regarding differences in basal responses between the active and passive responders. Though there does appear to be a discrepancy in PVN reactivity that coincides with a similar discrepancy in max speed (darting behavior), we feel that this cannot explain the effect only being seen in one group as both contain a vehicle and psilocin treated subgroup. **Text has been added to the discussion (pg. 13 line 18-22)** addressing this concern and highlighting the need for future work to examine such questions in greater detail.

3. Taken together "Psilocin increases activity in PVN which is associated with reduced active threat response". This is very cool! and maybe should be the title which is somewhat uninformative currently.

We thank the reviewer for their enthusiasm and for the appreciation of our findings. We agree that the increased PVN activity that is associated with reduced active threat response is a particularly compelling aspect of our work and have changed the title to **"Increased reactivity of the paraventricular nucleus of the hypothalamus and decreased threat responding in male rats following psilocin administration"** to better reflect the main findings of our study.

4. Thus, my feeling is that there are very strong results in this manuscript but that it is still difficult to uncover them with the many male and female comparisons and the repetitive nature of the writing (particularly the "Acute and prolonged effects of psilocin on PVN reactivity and behavior in males" then "...females" sections). I think this paper would be much stronger and perhaps ready for publication with just 4 figures if male and female data were combined for all experiments (except for corticosterone perhaps) -- Current graphs can just be cloned into the supplement, and combined graph presented as main figures.

We thank the reviewer again for their positive assessment of our 'strong results' and we have revised the manuscript so that all of the acute effects for both sexes are condensed into a single figure (**Figure 2**) and all of the prolonged effects are in a separate figure (**Figure 3**). This restructuring reduces repetitive language and increases the overall clarity of the data communication. We also considered the possibility of combining the data by sex, however, given the differences in baseline reactivity (see response to comment 2 above), we feel that maintaining the data communication separated by sex would be the most appropriate way to share these findings. This was in accordance with journal guidelines. We are not closed off to the idea of including a combined analysis in supplement and then showing data separated by sex in the figures, however, we feel that it may be perceived as redundant to do so and perhaps inappropriate given the differences in baseline that were found and are now reported in **Figure 1**.

1. I would like to thank the reviewers for the efforts made to address the concerns raised in the previous review. However, for this reviewer, the quality of data and conclusion drawn from the current study won't still merit the publication in Nat Comm.

We appreciate the reviewer's positive response to the edits made in our previous revision. We feel that particularly with the more in-depth analysis included in this secondary revision and the inclusion of new data on baseline sex differences in PVN reactivity (see response to Reviewer 1, comment 2, above), as well as the improved data presentation from this and the previous revision, the findings are of sufficient rigor and importantly are in support the conclusions drawn in a manner consistent with the publication standards of Nature Communications.

2. On the point of data quality, 1) for the c-Fos data, the authors argued with citation with some previous studies; however, the current studies showed extraordinary number of baseline c-Fos expression (I would guess each section would have more than 200 neurons positive). In addition, if any previous studies showing a low level of c-Fos at baseline, the current study should have that baseline expression of expression. Please see PMID 26412230 for the level of c-Fos expression at baseline at per section level. Bad experiments in previous publications couldn't justify this study here. Also the core data presented here solely relying on fiber photometry with a mild difference to support the conclusion are not robust.

The reviewer's concern regarding elevated levels of c-Fos expression compared to other studies is well received. Text had been previously added to acknowledge that injections are inherently stressful and therefore our baseline levels of c-Fos expression may not reflect basal activation. **Taking into consideration the concern of the reviewer, we have removed language referring to c-Fos expression as basal and instead refer to it as "stimulus-independent" (abstract: pg. 2 line 8; results: pg. 4 line 10; discussion: pg. 10 line 3, 12, 20).** We feel this is appropriate because c-Fos expression is being measured 2 hours prior to injection, and we feel that, by that point, injection induced elevations would be gone although we cannot rule out any lingering elevations as a result of injection stress. The current study provides valuable and novel insights into PVN function and a potential involvement of the PVN in the acute effects of psychedelic drugs that can be used to develop future studies investigating causal roles and circuit contributions.

3. On the significance part, all data presented here are based on correlation between activity and locomotion. There are no data that could show a causal role for PVN neurons in contributing the drug effects on behavior.

The reviewer brings up a fair point regarding a lack of causal inference in the current findings. Stress responding behavior is quite complex and could likely never be derived from a single brain region. While the inclusion of some sort of causal manipulation to alter the observed behavior would be an interesting and important finding, it lies outside the scope of the current work. The intention of these experiments was to examine the effects of the administration of psilocin on PVN reactivity and behavioral responding

following exposure to an aversive air-puff stimulus. Future work will explore a potential mediator of the observed effects. This had been previously addressed in the text based on prior reviewer comments, however **more text has been added in discussion (pg. 10 line 8-9).**

4. In addition, although it is true the psychedelic drugs produce some promising effects on psychiatric behaviors, none of the current experiments showed any effects on the beneficial effects by the used psychedelic drug. The significance of the changes in locomotion is not clear.

While the current work does not include any sort of “beneficial” or “therapeutic” effects of the psychedelic drug, we want to emphasize that this was intentional. When designing the experiment, we did not intend to model clinical conditions in adherence with specific NIH recommendations regarding preclinical psychedelic research (see NIMH guidelines: <https://grants.nih.gov/grants/guide/notice-files/NOT-MH-23-125.html>). We feel that the strength of rodent models is not to recapitulate complex human psychiatric disorders, but rather examine changes in neurobiology that may underly phenotypes associated with psychiatric disorders. Therefore, our approach was to examine neurophysiological changes in response to drug administration within a highly relevant brain region and, while we didn't include hallmark depressive-like/anxiety-like behavioral measures, we did include naturalistic behavior that could parallel stress coping strategies in humans. Here we show, for the second time (see Effinger et al. 2023), that active stress responding was predictive of drug sensitivity, which has potential predictive relevance in treatment responsiveness in human clinical populations. **This information has been added to the revised discussion (pg. 13 lines 25- pg. 14 line 1; pg. 15, lines 7-8).**

5. The data on the difference between males and females and between male aggressor and non-aggressor appear to be more interesting. Unfortunately no mechanistic studies were provided.

We appreciate the reviewer's shared enthusiasm for the intriguing findings of behavioral threat responding behavior and its potential involvement in drug sensitivity. While we do not have mechanistic/causal information regarding the observed effects of the drug on PVN function and behavior, we do feel that these findings are novel and important to the field. This work in conjunction with previously published work (Effinger et al., 2023) has inspired an entirely new line of research within the lab to focus on these behavioral stress-responding phenotypes to further understand the neurobiological substrates of this divergent behavior and the observed specificity of drug sensitivity. **Language has been added to the text (discussion: pg. 10 line 8-12, pg. 13 line 18-22) to address the limitations of the current work and to demonstrate the urgency to build on these findings and explore the leads uncovered by this body of work.**

Reviewer 3

1. In this revised manuscript, the authors have increased their ns for many of their experiments and moved many of the behavioral findings to the supplemental. The main

finding of the manuscript are that psilocin acutely increases PVN activity in response to an air puff in "active threat responding" male rats, but not passive threat responding male rats or female rats. Many of the findings are negative results. This is not necessarily a bad thing but is counter to mentions throughout the manuscript of the prolonged effects of psilocin on behavior and neuronal activity (i.e. days or weeks after administration). If psilocin only acutely shows a difference in PVN reactivity, it raises the question of the relevance of the PVN for psilocin's persistent effects on behavior.

The reviewer brings up a concern regarding the inclusion of negative results within our exploration of prolonged effects of psilocin. We have decided to include these data for three reasons.

1.) Psychedelics have been shown to reduce symptoms associated with several affective psychiatric disorders. Psilocybin was granted breakthrough therapy designation for treatment resistant depression, as it was shown to have potentially greater efficacy in symptom amelioration. One of the most compelling aspects of these compounds is the observation of prolonged improvements in treatment outcomes (months to even years out) following 1-2 doses of psilocybin in conjunction with psychotherapy. This rapid acting and persistent effect is not seen with the use of the current pharmacotherapies (i.e. SSRIs). Therefore, understanding the neurobiological mechanisms underlying the persistent effects may illuminate the therapeutic mechanisms underlying psychedelic drug action.

2.) Previously published work (Effinger et al. 2023) demonstrated prolonged decreases in central amygdala reactivity following a single dose of psilocin. This reduction (only seen in males) was seen 2-days following administration and remained as far as 28-days following administration. This effect was not seen in vehicle control or females. For this reason, we felt including prolonged time points was imperative.

3.) Lastly, we feel that excluding "negative" data is extremely detrimental to scientific advancement. I can personally speak to having done experiments and presented negative data at a conference and had someone approach me saying that they had done an experiment 30 years ago that demonstrated that our therapeutic compound had neurotoxic effects. Had this been published we would have saved an enormous amount of time and money. Therefore, we included our negative results to include the full extent of psilocin actions (and lack of actions) as both are important to the field at large. **Text has been added to discussion (pg. 12 line 22–25).**

2. The authors' responses to my previous comments are, for the most part, acceptable. The argument that the male-female increase in c-FOS staining in the PVN after psilocin injection seen in Fig. 1 reflects "basal" PVN activity is somewhat weak. The authors admit that injections themselves are stressors (lines 54, 55), so their c-FOS staining results are not measuring basal c-FOS levels. This changes the interpretation of their results: the stressful stimulus of psilocin injection increases PVN activity above that of the stressful stimulus of vehicle injection in both males and females. The change in PVN activity due to the air puff is due to a second stressful stimulus that seems to differentially affect males and females.

The reviewer's concern was also brought up by Reviewer 2 (question 1). The concern regarding elevated levels of c-Fos expression compared to other studies is well

received. Text had been previously added to acknowledge that injections are inherently stressful and therefore our baseline levels of c-Fos expression may not reflect basal activation. **Taking into consideration the concern of the reviewer, we have removed language referring to c-Fos expression as basal and instead refer to it as “stimulus-independent” (abstract: pg. 2 line 8; results: pg. 4 line 10; discussion: pg. 10 line 3, 12, 20).** We feel this is appropriate because c-Fos expression is being measured 2 hours prior to injection, and we feel that, by that point, injection induced elevations would be gone.” Beyond that, we feel that we have demonstrated a clear effect of psilocin on c-Fos expression given that both groups were tightly controlled such that the only difference between groups was the addition of psilocin.

On that note, text has been added and adjusted to address these concerns (**abstract: pg. 2 line 8; results: pg. 4 line 10; discussion: pg. 10 line 3, 12, 20**)

3. I would also recommend measuring changes in PVN reactivity using a two-way ANOVA versus just measuring the AUC or peak point. By eye, the female psilocin PVN response to an air puff in Fig. 3C looks to be overall greater than the female vehicle PVN response. It might not be statistically significant but I'd recommend checking.

This comment was a great addition to the paper. 2-way ANOVAs have been performed comparing binned mean traces (including data from each subject as opposed to just means and SEM). Wherever we found significant interactions or main effects, asterisks and/or inset text has been added to demonstrate 2-way ANOVA results in **Figure 1, 2, and 3**. While we agree there is absolutely a trend towards an effect of psilocin injection in the females, it did not reach significance and so is not included in the significant results.